# RANDLORA: FULL-RANK PARAMETER-EFFICIENT FINE-TUNING OF LARGE MODELS

**Paul Albert    Frederic Z. Zhang    Hemanth Saratchandran**
**Cristian Rodriguez-Opazo    Anton van den Hengel    Ehsan Abbasnejad**

Australian Institute for Machine Learning    The University of Adelaide

`{firstname.lastname}@adelaide.edu.au`
`https://github.com/PaulAlbert31/RandLoRA`

## ABSTRACT

Low-Rank Adaptation (LoRA) and its variants have shown impressive results in reducing the number of trainable parameters and memory requirements of large transformer networks while maintaining fine-tuning performance. The low-rank nature of the weight update inherently limits the representation power of fine-tuned models, however, thus potentially compromising performance on complex tasks. This raises a critical question: when a performance gap between LoRA and standard fine-tuning is observed, is it due to the reduced number of trainable parameters or the rank deficiency? This paper aims to answer this question by introducing RandLoRA, a parameter-efficient method that performs full-rank updates using a learned linear combinations of low-rank, non-trainable random matrices. Our method limits the number of trainable parameters by restricting optimization to diagonal scaling matrices applied to the fixed random matrices. This allows us to effectively overcome the low-rank limitations while maintaining parameter and memory efficiency during training. Through extensive experimentation across vision, language, and vision-language benchmarks, we systematically evaluate the limitations of LoRA and existing random basis methods. Our findings reveal that full-rank updates are beneficial across vision and language tasks individually, and even more so for vision-language tasks, where RandLoRA significantly reduces—and sometimes eliminates—the performance gap between standard fine-tuning and LoRA, demonstrating its efficacy.

## 1 INTRODUCTION

Large pre-trained models that leverage broad data have demonstrated significantly improved generalization capabilities and remarkable versatility across diverse tasks. However, the resultant high parameter count also leads to a significant increase in the computational resources required to fine-tune such models on downstream tasks. To tackle this issue, parameter-efficient fine-tuning (PEFT) approaches such as low-rank adaptation (LoRA) (Hu et al., 2022), draw inspiration from the low intrinsic dimensionality of pre-trained models (Li et al., 2018; Aghajanyan et al., 2021) and characterize the weight updates as the product of two low-rank matrices, substantially reducing the number of trainable parameters and memory requirements during training. This formulation leads to an adaptable number of trainable parameters, as one modifies the rank of the matrices, providing great flexibility under various resource constraints.

In spite of the strong performance of LoRAs in parameter-efficient settings, our investigation uncovers an accuracy plateau, wherein an increase of rank and thus learnable parameters fail to bridge the accuracy gap with standard fine-tuning. These undesirable scaling properties (Kopiczko et al., 2024) raise questions about the inherent limitations imposed by the low-rank structure, particularly when tackling complex tasks that benefit from larger parameter counts. This issue would ideally be addressed by introducing full-rank updates while maintaining the parameter-efficiency. To this end, we propose RandLoRA, a PEFT method that leverages a set of linearly-independent random bases in the form of non-trainable low-rank matrices. By solely learning scaling coefficients for the linear combination of the random low-rank bases, our method achieves full-rank updates, while maintain-

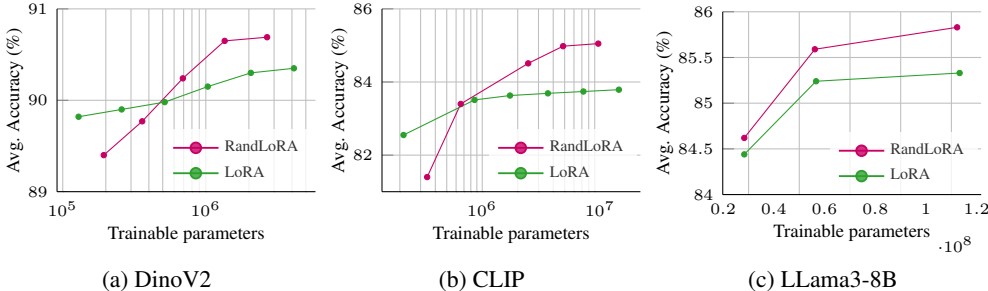

Figure 1: LoRA becomes limited by the rank of its update. We train DinoV2 and CLIP to classify 21 image datasets and LLama3-8B to solve 8 commonsense reasoning tasks.

ing low memory usage. As a result, RandLoRA strikes a balance between parameter efficiency and full-rank updates, allowing for more flexible and effective fine-tuning.

Through extensive experimentation, we empirically demonstrate the limitations of the low-rank formulation in LoRA, particularly on vision-language tasks, and show how RandLoRA can improve performance under similar parameter budget. Figure 1 summarizes our findings across pure vision (DinoV2), vision-language (CLIP) and commonsense reasoning (LLama3-8B), where increasing LoRA's parameter count has highly diminishing returns. We find that RandLoRA outperforms LoRA as the parameter budget expands, while remaining parameter efficient thanks to its full-rank update strategy. We conclude our investigation with an insightful discussion on the distinctive characteristics of RandLoRA where our analysis reveals that, in contrast to LoRA, RandLoRA yields activation patterns in deeper layers that closely align with those obtained through full fine-tuning. Furthermore, our visualization of the loss landscape reveals that the local minima reached by Rand-LoRA is often closer to that reached by standard fine-tuning, and it always leads to a lower loss than LoRA for an equal parameter count. Additionally, we explore the integration of sparse random bases, where initial findings highlight that sparse bases preserves the performance of RandLoRA. This suggests promising avenues to further reduce memory and computational requirements when training large transformer models, without compromising model performance.

Our contributions are summarized as:

1. We investigate the interplay between rank and number of trainable parameters when fine-tuning large pre-trained models, highlighting the limitations of LoRA in improving performance when larger ranks are required.

2. We propose RandLoRA, a novel parameter-efficient fine-tuning (PEFT) strategy based on random basis combinations, enabling full-rank updates without memory overhead over LoRA.

3. We rigorously assess RandLoRA across diverse pre-trained architectures and tasks, spanning pure vision and vision-language image classification to commonsense reasoning, demonstrating its versatility and effectiveness.

## 2 RELATED WORK

### 2.1 LOW RANK ADAPTATION OF LARGE MODELS

Low Rank Adaptation (LoRA) of large language models has revolutionized the fine-tuning paradigm, enabling memory-constrained adaptation to specialist tasks and democratizing access to larger models. Initially introduced by (Hu et al., 2022), LoRA leverages the observation that weight updates during fine-tuning can converge to suitable performances without necessitating full rank updates. By factorizing weight updates into the product of two low rank matrices, LoRA achieves a memory-efficient solution for adapting large models. Moreover, once the low rank matrices are

merged into the original weight matrix size, no latency is present during inference. Several improvements have been proposed to build upon LoRA's success. Weight-decomposed LoRAs (DoRA) (Liu et al., 2024) proposes to improve convergence by decomposing LoRA updates into magnitude and direction components. AdaLoRA (Zhang et al., 2023) and AutoLoRA (Zhang et al., 2024c), utilize specialized metrics or meta-learning to propose rank-adapted LoRA formulations that dynamically adjust the rank to suit every layer's need. Other improvements include initialization strategies for the low rank matrices using the truncated SVD of the pre-trained weights and where the whole decomposition is fine-tuned as in Pissa (Meng et al., 2024) or where only the singular value matrix is as in SVFT (Lingam et al., 2024) or LoRA-XS (Bałazy et al., 2024). Further improvements are proposed in HydraLoRA (Tian et al., 2024) where the scaling-up matrix of the low rank decomposition is split into multiple ones with a routing layer added to select the contribution of each head. This formulation enhances multi-task learning at the cost of losing the merging capabilities of LoRA in the pre-trained weight at test-time. These advancements collectively enhance the efficiency of LoRA, solidifying its position as a cornerstone of large language model fine-tuning.

## 2.2 Parameter-Efficient fine-tuning (PEFT) using Random Bases

Recent research has focused on further reducing the trainable parameter count of LoRA, a crucial aspect for low-shot applications where minimizing trainable parameters can prevent overfitting and enhance generalization. A promising direction involves utilizing random bases combinations, where randomly generated matrices are combined using a limited number of trainable parameters to estimate a weight update.

PRANC (Nooralinejad et al., 2023) pioneered the random base strategy by learning a weighted averaged of random matrices through back-propagation. PRANC's solution averages multiple full size weight matrices for each layer, leading to high memory consumption. To address this, the authors generate random bases on the fly during forward and backward passes using a fixed seed random number generator, reducing memory usage to that of the largest trained layer in the network at the cost of training latency.

Building upon PRANC, NOLA (Koohpayegani et al., 2024) introduces an improved algorithm where random bases are estimated as the product of two low-rank random matrices, each weighed using a learnable scalar and summed before matrix multiplication. This approach effectively approximates a rank 1 LoRA with significantly fewer trainable parameters and largely reduces memory consumption during training over PRANC.

Concurrently, VeRA (Kopiczko et al., 2024) proposed an alternative strategy utilizing a single high-rank random matrix (typically 256 or 1024), instead of summing multiple rank 1 matrices as in NoLA. VeRA also employs a scaling strategy of random bases distinct from NoLA, detailed in section 4, which relates to our approach. Both NOLA and VeRA achieve comparable performance to LoRA in few-shot fine-tuning scenarios while training substantially fewer parameters.

## 2.3 Alternative strategies for parameter-efficient fine-tuning

We report here on alternatives to weight tuning for parameter-efficient adaptation, specifically focusing on prompt tuning. Context Optimization (CoOP) (Zhou et al., 2022b) introduced learnable context vectors for CLIP class names, later generalized to instance-specific prompts in Conditional CoOP (CoCoOP) (Zhou et al., 2022a). Recent prompt tuning methods, like DePT (Zhang et al., 2024b) and PromptSRC (Khattak et al., 2023b), emphasize knowledge preservation by isolating shared subspaces or regularizing prompts. While parameter-efficient, prompt tuning can struggle with generalization beyond few-shot settings (Han et al., 2024) and may be less effective than LoRA as data increases (Zanella & Ben Ayed, 2024). We therefore consider prompt tuning orthogonal to weight-tuning for the scope of this paper and exclude it from direct RandLoRA comparisons except for early results found in Appendix B.3.

## 3 Motivations

Our literature review reveals that research on improving LoRA is focused on reducing the number of trainable parameters further, either through adaptable ranks or by using fixed or shared low rank

projection matrices. When looking at moderate to larger parameter budgets however LoRA remains highly competitive.

We identify that early research has convincingly demonstrated the promise of random basis combinations as a parameter-efficient strategy for large models, particularly in few-shot scenarios. Two approaches have emerged, each representing a distinct paradigm. VeRA advocates for a unique random base with large rank, while NoLA proposes to average a large number of random bases with small ranks. Both approaches report performance comparable to LoRA in few-shot scenarios while converging on a significantly reduced number of trainable parameters. However, as we will demonstrate, this reduction comes at the cost of limited performance when venturing beyond few-shot learning, limiting the scalability of these algorithms.

Finally, we report that LoRA is predicated on the assumption that low-rank updates suffice for fine-tuning large models. We aim in this paper to question the universality of this hypothesis, exploring scenarios where full rank alternatives may be necessary. The fundamental question follows: is parameter efficiency achieved through low-rank approximation limited by (1) the low-rank nature of the update or (2) by the low parameter count. Can parameter-efficient full rank updates provide a more accurate solution ? This paper aims to address these questions, exploring the balance between parameter efficiency and low-rank fine-tuning of large transformer models, and shedding light on the limitations of existing approaches.

## 4 RANDLoRA—PARAMETER-EFFICIENT FINE-TUNING WITH FULL RANK

### 4.1 WEIGHT UPDATES AS A SUM OF LOW-RANK MATRICES

Let $W_0 \in \mathbb{R}^{D \times d}$ be a weight matrix of a large pre-trained model. Fine-tuning aims to find an appropriate $\Delta W \in \mathbb{R}^{D \times d}$, such that the fine-tuned weights $W_0 + \Delta W$ lead to an adapted model, tailored to a specific downstream task. Without loss of generality, let us assume $d < D$. The motivation behind RandLoRA stems from the singular value decomposition (SVD) of $\Delta W$, i.e., $\Delta W = U\Sigma V^{\mathsf{T}}$, where $U \in \mathbb{R}^{D \times d}$, $\Sigma \in \mathbb{R}^{d \times d}$, $V \in \mathbb{R}^{d \times d}$. This decomposition can be written as the sum of the product of rank-one matrices, as follows

$$\Delta W = \sum_{i=1}^{d} \mathbf{u}_i \sigma_i \mathbf{v}_i^{\mathsf{T}}, \tag{1}$$

where $\mathbf{u}_i$ and $\mathbf{v}_i$ denote the columns of $U$ and $V$, respectively. We suggest that in this context, low-rank updates such as LoRAs can be characterized as an approximation of the few largest singular values while the rest of the information in $\Delta W$ being discarded. To better illustrate this point, let us denote the rank of LoRA by $r$ and for brevity of exposition, assume $d$ is divisible by $r$. We rewrite equation 1 as a sum of the product of rank-$r$ matrices, as follows

$$\Delta W = \sum_{j=1}^{n} U_j \Sigma_j V_j^{\mathsf{T}}, \tag{2}$$

where $U_j \Sigma_j V_j^{\mathsf{T}} = \sum_{i=rj}^{r(j+1)} \mathbf{u}_i \sigma_i \mathbf{v}_i^{\mathsf{T}}$ and where $n = d/r$. This formulation reveals how LoRA models the approximates the first low-rank partition $U_1 \Sigma_1 V_1^{\mathsf{T}}$, and implicitly assumes $\sum_{j=2}^{n} U_j \Sigma_j V_j^{\mathsf{T}} \approx 0$. We however argue that the remaining $n-1$ terms can play a crucial role when capturing more complex task-specific variations that require larger deviations from the pre-trained weight $W_0$.

### 4.2 PARAMETER-EFFICIENT APPROXIMATION OF LOW-RANK MATRICES

Approximating more terms in the decomposition of $\Delta W$ using LoRA's formulation quickly becomes parameter inefficient, culminating to $Dd + d^2$ parameters for a full rank $d$ in place of the original $Dd$ parameters of $\Delta W$. To perform full-rank updates while maintaining parameter-efficiency, we propose instead to approximate each term of $\Delta W$ in equation 2 using low-rank random bases where only scaling coefficients are learned,

$$\Delta W = \sum_{j=1}^{n} B_j \Lambda_j A_j \Gamma_j, \tag{3}$$

where $B_j \in \mathbb{R}^{D \times r}$ and $A_j \in \mathbb{R}^{r \times d}$ are non-trainable, random matrices. The two learnable diagonal scaling matrices, $\Lambda_j \in \mathbb{R}^{r \times r}$ and $\Gamma_j \in \mathbb{R}^{d \times d}$ are unique to each of the $n$ terms and fulfill complementary roles to improve the approximation. We aim for $A_j\Gamma_j$ transform the input features into an low-dimensional space (rank-$r$), $\Lambda_j$ to scale the compressed features which are then transformed back into the desired output space by $B_j$.[1] Since $\Gamma_j$ operates on the column space of $A_j$ and is unique to each $A_j$, we use a unique shared matrix $A \in \mathbb{R}^{r \times d}$ across all $n$ terms without loss of expressivity but reducing memory consumption. With a shared $A$, we formulate the update as

$$\Delta W = \sum_{j=1}^{n} B_j \Lambda_j A \Gamma_j. \tag{4}$$

To achieve a full-rank update, we set $n = d/r$, leading to $\frac{d}{r}(d + r) = d^2/r + d$ learnable parameters. Note that unlike LoRA, the number of learnable parameters is inversely proportional to the rank of the random bases in RandLoRA, as increasing the rank of the bases leads to a reduction in trainable parameters while maintaining full rank. In summary, RandLoRA trades-off approximation accuracy for scope, sacrificing a more precise representation of the individual SVD elements of $\Delta W$ to capture a larger portion of its singular value decomposition.

### 4.3 CONVERGENCE ANALYSIS

In this section, we present a theorem showing that weight updates using RandLoRA is an accurate approximation of general matrices under certain theoretical conditions.

**Theorem 4.1.** *Let $W$ be a fixed $D \times d$ matrix, with $D > d$ and rank$(W) = d$. Fix $1 \le n \le d$, such that $d = nr$. The matrix $W$ can be factorized using SVD as*

$$W = \sum_{j}^{n} U_j \Sigma_j V_j^{\mathsf{T}}, \tag{5}$$

*where $U_j \in \mathbb{R}^{D \times r}$, $V_j \in \mathbb{R}^{r \times d}$ are partitions of the left and right singular vectors, and $\Sigma_j \in \mathbb{R}^{r \times r}$ contains r singular values. For each $1 \le j \le n$, let $B_j$ denote a random $D \times r$ matrix whose entries are drawn i.i.d from either a Gaussian or uniform distribution, $A_j$ denotes an $r \times d$ matrix whose entries are drawn similarly, $\Lambda_j$ is a diagonal $r \times r$ matrix and $\Gamma_j$ is a diagonal $d \times d$ matrix drawn similarly. Assume*

$$\|U_j \Sigma_j V_j^{\mathsf{T}} - B_j \Lambda_j A_j \Gamma_j\|_F \le \epsilon \tag{6}$$

*for each $1 \le j \le n$ for some $0 < \epsilon$. Then we have that with probability 1 that each $B_j \Lambda_j A_j \Gamma_j$ has full rank and*

$$\left\| W - \sum_{j=1}^{n} B_j \Lambda_j A_j \Gamma_j \right\|_F \le n \cdot \epsilon. \tag{7}$$

For details on the proof of theorem 4.1 please refer to appendix D.1.

Theorem 4.1 is premised on $B_j \Lambda_j A_j \Gamma_j$ being a good approximation for the $r$-truncated singular value of $\Delta W$, which is shown to be true empirically in VeRA (Kopiczko et al., 2024) for example. We show in this case that $\Delta W$ can be accurately approximated as $\sum_{j=1}^{n} B_j \Lambda_j A_j \Gamma_j$, motivating RandLoRA's formulation. In contrast, since the best approximation a rank-$r$ LoRA can achieve is the $r$-truncated SVD of $W$, then by Eckart-Young-Mirsky theorem, the Frobenius norm of the difference between $W$ and low-rank adaptation $BA$ is lower bounded as follows

$$\|W - BA\|_F \ge \left\| W - \sum_{i=1}^{r} \mathbf{u}_i \sigma_i \mathbf{v}_i^{\mathsf{T}} \right\|_F = \sum_{i=r+1}^{d} \sigma_i^2. \tag{8}$$

We conclude that while LoRA's rank $r$ approximation is limited by the sum of the last $d - r - 1$ squared singular values of $W$, RandLoRA does not present this low bound and is only limited by how close ($\epsilon$) can $B_j \Lambda_j A_j \Gamma_j$ approximate length-r segments of the SVD of $W$.

---

[1]The formulation of our method is similar to that of VeRA (Kopiczko et al., 2024), which will be discussed in detail in section 6.5.

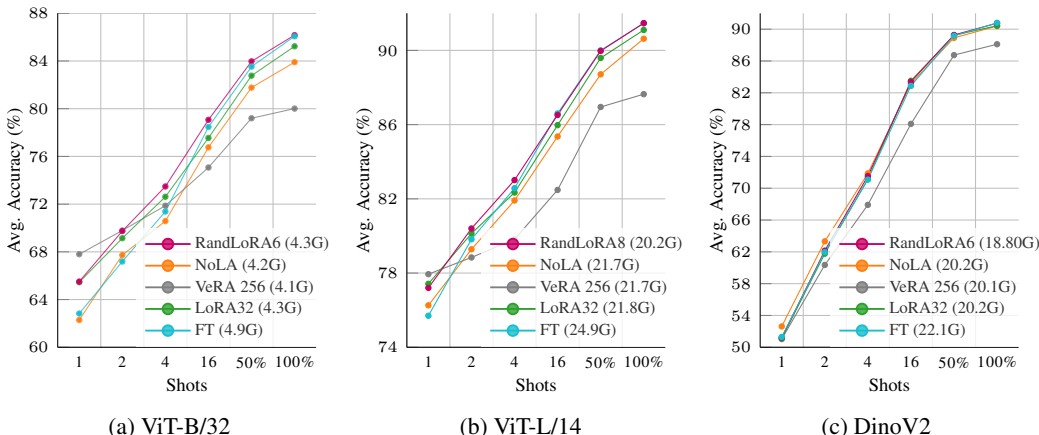

(a) ViT-B/32        (b) ViT-L/14        (c) DinoV2

Figure 2: Tuning CLIP and DinoV2 vision encoders for image classification. Accuracy averaged over 21 datasets. We additionally report max GPU VRAM usage during training.

# 5 EXPERIMENTS

## 5.1 EXPERIMENTAL SETTINGS

We conduct a comprehensive comparison with three state-of-the-art approaches: LoRA (Hu et al., 2022), NoLA (Koohpayegani et al., 2024), and VeRA (Kopiczko et al., 2024). We perform a hyperparameter search to identify optimal settings for LoRA, NoLA, VeRA, and RandLoRA to ensure a fair comparison. More details about the experimental settings can be found in appendix C. Additional experiments on the General Language Understanding Evaluation (GLUE) (Wang et al., 2019) and End-to-end (E2E) Novikova et al. (2017) natural language generation benchmarks as well as further comparison with prompt-tuning algorithms are available in appendix B.

## 5.2 VISION: DINOV2 AND CLIP'S VISION BACKBONE

We evaluate fine-tuning vision backbones for image classification using pre-trained ViT-B/14 DinoV2 (Oquab et al., 2023) and ViT-B/32, ViT-L/14 CLIP (Radford et al., 2021) vision only backbones. We fine-tune on 21 datasets (Appendix C.1, Table 7) and evaluate {1, 2, 4, 16}-shot learning and performance with 50% and 100% training data.

We compare RandLoRA to LoRA rank 32 where RandLoRA's rank is adjusted to match LoRA's parameters, and include VeRA and NoLA as random base alternatives. We fine-tune the vision backbones and learn linear classifiers for DinoV2, or use frozen CLIP language embeddings for classification. Results are displayed in Figure 2 where we also report VRAM usage, detailed results are available in Appendix E.2.

We find that LoRA exhibits a smaller accuracy gap with standard fine-tuning (FT) on DinoV2 than CLIP. With equal parameters, RandLoRA improves over LoRA, bridging the FT gap in both cases. We believe that LoRA's success on the DinoV2 backbone is partly explained by its training objective (see Section 6.1). RandLoRA demonstrates LoRA's rank limitation for CLIP architectures and the benefit of full-rank updates in matching FT performance. VeRA and NoLA are efficient in few-shot settings but become limited with more data.

## 5.3 VISION-LANGUAGE: CLIP

We extend in this section our experimental setting to fine-tuning CLIP-like transformer architectures on classification datasets where contrary to section 5.2 both the language and vision encoders of CLIP are trained. We add ImageNet (Krizhevsky et al., 2012) to the dataset pool to scale up to 22 classification datasets. To assess the effectiveness of RandLoRA compared to LoRA on models of varying sizes, we consider three variants of pre-trained CLIPs from the open-clip repository (Cherti et al., 2023): ViT-B/32 (151M parameters), ViT-L/14 (428M parameters) and ViT-H/14 (1B pa-

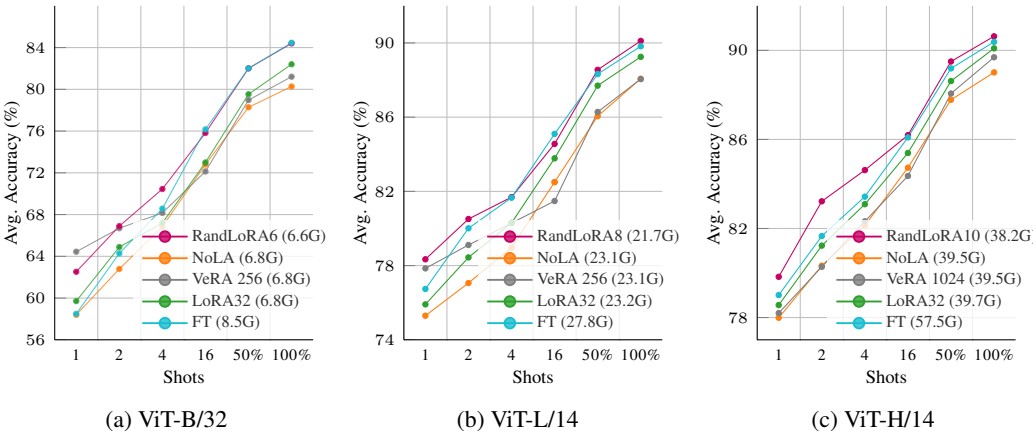

Figure 3: Tuning CLIP's vision and language encoders for image classification. Accuracy averaged over 22 datasets. We additionally report max GPU VRAM usage during training.

rameters). We scale the rank of the random bases in RandLoRA in the same way as section 5.2 to maintain a number of parameters comparable to a rank 32 LoRA: RandLoRA-{6,8,10} for ViT-{B/32,L/14,H/14} respectively.

A summary of results is available in Figure 3 with detailed results being available in appendix E.1. Because fine-tuning vision-language architectures such as CLIP is a harder optimization problem, we observe the existence of a larger performance gap between full fine-tuning and LoRA than for pure vision, which we confirm is not bridged by increasing the rank of LoRA (see Figure 1). This suggests that increasing parameter count is not enough, pointing towards the rank of the update as the possible limit to the performance of LoRA. When running RandLoRA with the same amount of trainable parameters, we observe that the gap with fine-tuning is bridged. When compared with NoLA and VeRA we come to the same conclusions as section 5.2 although VeRA is this time much more competitive for larger data budgets, hinting towards the importance of high ranks for finetuning CLIP-like vision language architectures. We also report that our base sharing strategy allows RandLoRA to decrease VRAM usage over LoRA which can be relevant for large architectures such as ViT-H/14.

## 5.4 COMMONSENSE REASONING

We evaluate RandLoRA for fine-tuning LLMs on eight commonsense reasoning tasks (see Appendix C.4). We fine-tune Qwen2 (0.5B), Phi3 (3B), and Llama3 (8B) models and assess data efficiency by training on both a 170,000-sample full dataset and a 15,000-sample subset, following Hu et al. (2023).

Table 1 compares RandLoRA to LoRA, VeRA, and NoLA. We test two LoRA ranks: rank-16 ("Efficient") and rank-32 ("Performant"). We then scale RandLoRA the same or lower amount of parameters to ensure a fair comparison. Detailed results are found in Appendix 15

RandLoRA performs competitively with, and sometimes surpasses, LoRA. Phi3's strong zero-shot abilities enable VeRA and NoLA to achieve strong results despite fewer parameters. Conversely, Qwen2 and Llama3 require more adaptation, challenging VeRA and NoLA to match LoRA's performance. The 15k-sample regime can lead to overfitting when scaling trainable parameters for LoRA and RandLoRA, decreasing performance even with dropout regularization. When training on the full 170k samples, RandLoRA consistently outperforms LoRA. Results comparing with DoRA (Liu et al., 2024) for LLama3 only are available in Table 6 in the appendix where RandLoRA outperforms both DoRA and LoRA for larger parameter budgets, while DoRA and LoRA are competitive at "Efficient" budgets. We conclude RandLoRA is a compelling alternative to LoRA and DoRA for LLM fine-tuning, especially with larger datasets and parameter budgets.

Table 1: Parameter-efficient fine-tuning of Large Language Models (LLMs). Results averaged over 8 commonsense reasoning tasks. We bold the best accuracy between parameter-equivalent RandLoRA and LoRA configurations.

| Network | Size | ZeroShot | NoLA | VeRA | LoRA | | RandLoRA | |
| --- | --- | --- | --- | --- | --- | --- | --- | --- |
| | | | | | Efficient | Performant | Efficient | Performant |
| Qwen2-0.5b | 15k | 5.2 | 42.6 | 48.1 | 53.2 | 52.3 | **53.5** | 52.9 |
| | 170k | 5.2 | 47.4 | 51.8 | 57.4 | 57.3 | 57.7 | **57.9** |
| Phi3-3b | 15k | 65.4 | 80.4 | 78.6 | 81.8 | 80.3 | 81.7 | **82.3** |
| | 170k | 65.4 | 82.3 | 81.4 | 84.6 | 85.0 | 84.7 | **85.2** |
| LLama3-8b | 15k | 27.0 | 76.9 | 77.1 | 82.7 | **83.1** | 81.0 | 81.3 |
| | 170k | 27.0 | 81.2 | 81.7 | 84.4 | 85.2 | 84.6 | **85.6** |

Figure 4: How close do RandLoRA and LoRA get to standard fine-tuning ? We compare CKA scores of RandLoRA and LoRA with fine-tuned activations (top) and the mode connectivity in the loss landscape of UCF101 (bottom)

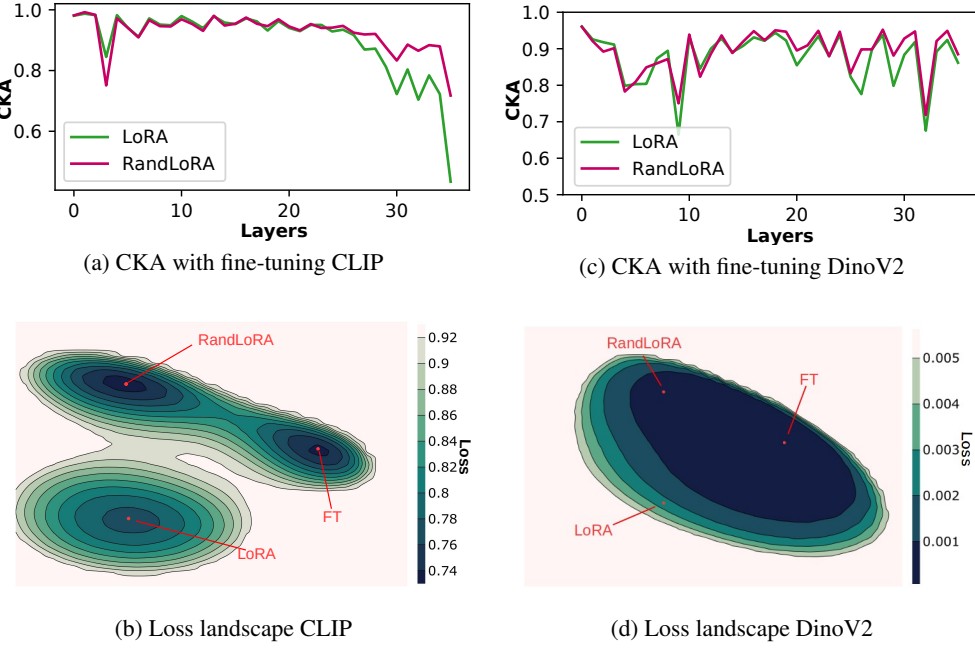

(a) CKA with fine-tuning CLIP

(c) CKA with fine-tuning DinoV2

(b) Loss landscape CLIP

(d) Loss landscape DinoV2

## 6 DISCUSSION

### 6.1 SIMILARITIES WITH FINE-TUNING: ACTIVATIONS

We evaluate activation similarity to assess LoRA and RandLoRA's ability to mimic fine-tuned model activations. Using the Centered Kernel Alignment (CKA) (Kornblith et al., 2019) metric, we measure the similarity between activations of LoRA, RandLoRA, and a fully fine-tuned model. This protocol assesses how well each method captures dataset-specific activation patterns. Figure 4a shows CKA scores for self-attention and MLP layers in CLIP and DinoV2 vision backbones, averaged over 5 datasets where RandLoRA imrpoves over LoRA. For CLIP, LoRA's CKA decreases in deeper layers, losing alignment with fine-tuned activations. RandLoRA, with equal parameters, matches LoRA's early layer alignment but improves upon it in deeper layers. This CKA drop for LoRA in deeper layers is absent in DinoV2, explaining LoRA's near-identical accuracy to fine-tuning on DinoV2. This difference likely arises from training objectives: DinoV2's visual objective creates classification-ready features needing minimal weight adjustments, thus low-rank LoRA suf-

Table 2: Ablation on the rank of the updates. The same amount of trainable parameters is used in all methods.

| Method | Rank | Accuracy |
|---|---|---|
| LoRA | 32 | 83.74 |
| RandLoRA-a | 32 | 83.62 |
| RandLoRA-b | 384 | 85.32 |
| RandLoRA-6 | 768 | 85.98 |

Table 3: Fine-tuning CLIP or LLama3 using Rand-LoRA different random distributions or base sparsity.

| Model | Sparsity | Accuracy |
|---|---|---|
| CLIP-ViT-B/32 - uniform | 0% | 85.98 |
| CLIP-ViT-B/32 - normal | 0% | 85.61 |
| CLIP-ViT-B/32 - binary | 0% | 85.52 |
| CLIP-ViT-B/32 | 66% | 85.43 |
| CLIP-ViT-B/32 | 93% | 85.57 |
| CLIP-ViT-B/32 | 98% | 84.35 |
| CLIP-ViT-B/32 | 99% | 83.34 |
| LLama3-8b | 0% | 85.59 |
| LLama3-8b | 66% | 85.42 |

fices. CLIP's multimodal objective, however, demands higher ranks for effective adaptation to vision tasks.

## 6.2 SIMILARITIES WITH FINE-TUNING: LOSS LANDSCAPE

We analyze loss landscape connectivity for models fine-tuned with standard fine-tuning, LoRA, and RandLoRA. We visualize a 2D loss landscape plane by positioning LoRA, RandLoRA, and fine-tuning models at (0,0), (1,0), and (0.5,1) respectively. For each point $(x, y)$ on this plane, we interpolate model weights by solving for coefficients $\alpha_i$ (where $\sum_{i=1}^{3} \alpha_i = 1$) and evaluate the interpolated model's loss on a $5\%$ training subset. Figure 4b shows that for CLIP, RandLoRA reaches a deeper loss minima than LoRA, often with a low-loss path to the fine-tuning optimum, and despite training the same parameter count. For DinoV2, all optima reside in a shared low-loss basin, with LoRA already close to fine-tuning, reflecting LoRA's strong performance on this task. These visualizations reinforce LoRA's low rank it particularly limiting for complex tasks, and demonstrate RandLoRA's ability to achieve deeper minima than LoRA with equal parameters due to full-rank updates. Appendix A provides 3D visualizations for additional datasets.

## 6.3 FURTHER STUDIES ON FULL VS LOW RANK FINE-TUNING OF CLIP

We investigate whether RandLoRA's CLIP performance advantage over LoRA stems from better SVD approximation or its full-rank capability. We ablate RandLoRA with two rank-controlled variants. RandLoRA-a restricts the update rank to $r$ by averaging bases before multiplication: $\Delta W = \left( \sum_{i=1}^{N} B_i \Lambda_i \right) \left( \sum_{i=1}^{N} A_i \Gamma_i \right)$. RandLoRA-b uses half-rank updates by setting $N = \mathrm{rank}(\Delta W)/r/2$ and adjusting base rank to maintain parameter count parity with RandLoRA-$r$. All variants train the same parameters, only update rank varies. Table 2 presents accuracy on 100% of 22 datasets for CLIP ViT-B/32. Results show that higher update rank correlates with better performance, given equal parameter counts. This supports the importance of large rank updates, particularly for CLIP fine-tuning.

## 6.4 SPARSE RANDOM MATRICES

We propose to investigate using sparse random matrices for improved memory and computational efficiency, drawing inspiration from random projection literature and the Johnson-Lindenstrauss lemma (Lindenstrauss & Johnson, 1984). We adopt the sparse construction from Bingham & Mannila (2001) and Li et al. (2006), where matrix elements are $\{-1, 0, 1\}$ with probabilities $\{\frac{1}{s}, 1-\frac{2}{s}, \frac{1}{s}\}$ ($s \in [2, \sqrt{D}]$ for $W \in \mathbb{R}^{D \times d}$), followed by normalization. Appendix C.6 discusses why this formulation preserves full rank. Table 3 shows experimental results using these sparse bases in RandLoRA. We explore sparsity ratios $s \in \{2, 6, \sqrt{D}, 100, 200\}$, achieving sparsity levels from 66 to 99%. Consistent with Li et al. (2006), the recommended sparsity levels ($\sqrt{D}$) yield performance comparable to dense matrices, theoretically reducing memory and compute. However, higher sparsity can de-

grade accuracy, suggesting potential for optimized RandLoRA variants using compute-optimized sparse random bases.

## 6.5 SUMMARY OF DIFFERENCES WITH RELATED RANDOM BASES ALGORITHMS

Prior work like VeRA (Kopiczko et al., 2024) and NoLA (Koohpayegani et al., 2024) utilizes random bases for parameter-efficient fine-tuning. However, unlike VeRA and NoLA which approximate a low-rank LoRA update, RandLoRA aims to approximate the full-rank weight update. It could be argued that VeRA approximates only the first block in a decomposition of $W$, whereas RandLoRA approximates all blocks. Thus, while VeRA and NoLA improve parameter-efficiency while maintaining low-rank updates, RandLoRA addresses cases requiring full-rank updates. Furthermore, Equation equation 4 evidences the flexibility in RandLoRA's parameter count, ranging from VeRA's parameter efficiency ($r = \text{rank}(W)$) to full fine-tuning parameters ($r = 1$) while maintaining full-rank.

## 6.6 LIMITATIONS

Despite RandLoRA's effectiveness, we identify three key limitations for future research.

First, RandLoRA introduces computational overhead in weight update calculations, increasing training time for larger models (Appendix C.6.1). We however evidence room for improvement using ternary sparse bases in Section 6.4. Future work should explore matmul-free matrix combinations using these ternary sparse bases. Efficient implementations could replace costly matrix products with simple aggregations, eliminating floating-point arithmetic (Li et al., 2006), and accelerating RandLoRA training time pending the development of optimized CUDA kernels (Zhu et al., 2024).

Second, exploring non-random, optimal bases $B_i$ and $A$ could improve convergence and efficiency by further reducing $\epsilon$ in equation equation 6. Discovering such bases, potentially through experiments or decomposition of pre-trained weights (Bałazy et al., 2024; Meng et al., 2024), is a promising research direction to enhance RandLoRA.

Third, hybrid approaches combining LoRA and RandLoRA warrant investigation. LoRA could estimate the dominant SVD components of $W$, while RandLoRA captures the remaining spectral information efficiently. Despite challenges in harmonizing training objectives, a starting point would use RandLoRA to refine a LoRA when convergence is insufficient. Addressing these limitations will further improve RandLoRA's potential for efficient full-rank fine-tuning.

## 7 CONCLUSION

This paper introduces RandLoRA, a method achieving parameter efficiency and low memory cost while enabling full rank model updates. Our findings underscore the critical importance of full-rank updates when fine-tuning pre-trained architectures and we observe that our approach surpasses LoRA's performance for an equal parameter count, highlighting the value of full-rank updates in large model fine-tuning. Through extensive experiments across diverse tasks we demonstrated the efficacy of our method. While RandLoRA incurs additional computational overhead due to random basis multiplications, memory consumption remains contained and we provide venues for reducing this compute in practice. As a results, RandLoRA offers a viable alternative to LoRA for fine-tuning large pre-trained models on consumer-grade hardware. Our results have significant implications for efficient and effective model adaptation, prompting for future research in scalable and versatile full-rank fine-tuning techniques.

## ACKNOWLEDGMENTS

This research is funded in part by the Australian Government through the Australian Research Council (Project DP240103278), and the Centre of Augmented Reasoning at the Australian Institute for Machine Learning, established by a grant from the Department of Education. This work is also supported by supercomputing resources provided by the Phoenix HPC service at the University of Adelaide.

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
