

(a) CIFAR-100        (b) Food-101        (c) UCF-101

Figure 5: Mode connectivity in the loss landscape when tuning CLIP for image classification. Interactive 3D figures are available in the supplementary material

## A   3D VISUALIZATIONS OF CLIP'S LOSS LANDSCAPE

We propose here further visualizations of the mode connectivity between LoRA, RandLoRA and standard fine-tuning. To compute the loss value between the minimas reached by LoRA, RandLoRA and fine-tuning, define a 2D plane using 3 equidistant points representing LoRA, standard fine-tuning and RandLoRA and we then solve for interpolation coefficients $\alpha_{1..3}$ so that their sum equals 1. The weights of the model we evaluate is then $W_0 + \alpha_1 \text{LoRA} + \alpha_2 \text{FT} + \alpha_3 \text{RandLoRA}$. The loss is evaluated on a fixed $5\%$ subset of the training set. Since the process of evaluating the loss at all coordinates on the plane is time consuming, we only perform this study for the CLIP-ViT-B/32 architecture where RandLoRA is especially successful. In all visualizations, the number of trainable parameters for LoRA and RandLoRA are the same. We clamp loss values $20\%$ above the shallowest minima to improve visualization. 3D representation as well as the associated 2D elevation projection is provided in Figure 5. The interactive 3D figures are provided in the HTML format in the supplementary material.

## B   ADDITIONAL RESULTS

KroneckerWe report here further results on the General Language Understanding Evaluation (GLUE) (Wang et al., 2019) and End-to-end (E2E) (Novikova et al., 2017) generation benchmarks. While GLUE is a text classification task, E2E is a natural language generation task. We also report results comparing RandLoRA and LoRA with a prompt tuning baseline (Zhang et al., 2024b) for classification using CLIP's vision backbone as in section 5.2 in appendix B.3

### B.1   GLUE RESULTS

We report results for RandLoRA and compare with LoRA and VeRA on the SST-2, MRPC, COLA, QNLI, RTE and STS-N tasks. We report Matthew's correlation for CoLA, Pearson correlation for STS-B, and accuracy for the remaining tasks. We report results using the RoBERTa network Liu et al. (2019) in the base and large configurations and perform 5 runs to report average performance and one standard deviation. Results are displayed in Table 4. We find that for the smaller RoBERTa-base architecture (125M parameters), all algorithms reach the same performance. For the larger RoBERTa-large variant (355M parameters), a larger gap is observed where RandLoRA improves over the performance of VeRA and LoRA. These findings are in line with the experiments in the main body of the paper where we find that RandLoRA provided larger improvements for larger models in Figure 3.

### B.2   E2E RESULTS

We train RandLoRA and LoRA on the E2E dataset using the GPT-2 medium architecture (Radford et al., 2019) (355M parameters).

Table 4: Results on GLUE datasets with the RoBERTa-base and RoBERTa-large models.

| | | | | RoBERTa-base | | | | |
|---|---|---|---|---|---|---|---|---|
| Method | Params | SST-2 | MRPC | COLA | QNLI | RTE | STS-N | Average |
| VeRA-1024 | 0.26M | $91.9 \pm 0.4$ | $88.4 \pm 1.2$ | $59.9 \pm 2.2$ | $90.5 \pm 0.4$ | $74.9 \pm 1.5$ | $90.4 \pm 0.2$ | $82.7 \pm 0.3$ |
| LoRA-4 | 0.7M | $94.4 \pm 0.5$ | $87.3 \pm 0.2$ | $58.4 \pm 0.8$ | $92.7 \pm 0.2$ | $71.5 \pm 1.2$ | $90.5 \pm 0.1$ | $82.4 \pm 0.3$ |
| RandLoRA-64 | 0.7M | $92.2 \pm 0.3$ | $88.0 \pm 1.5$ | $59.4 \pm 2.1$ | $91.3 \pm 0.4$ | $74.7 \pm 1.9$ | $90.3 \pm 0.2$ | $82.6 \pm 0.5$ |
| | | | | RoBERTa-large | | | | |
| VeRA-256 | 0.26M | $95.8 \pm 0.3$ | $89.3 \pm 1.2$ | $65.3 \pm 1.1$ | $94.1 \pm 0.3$ | $81.6 \pm 0.8$ | $91.8 \pm 0.1$ | $86.3 \pm 0.3$ |
| LoRA-4 | 1.8M | $95.5 \pm 0.2$ | $87.2 \pm 0.7$ | $64.7 \pm 1.2$ | $94.5 \pm 0.1$ | $83.6 \pm 0.4$ | $91.8 \pm 0.1$ | $86.2 \pm 0.3$ |
| RandLoRA-100 | 1.8M | $95.5 \pm 0.3$ | $90.1 \pm 0.4$ | $67.4 \pm 0.3$ | $94.1 \pm 0.3$ | $84.5 \pm 0.3$ | $91.4 \pm 0.6$ | $87.2 \pm 0.1$ |

### B.3 COMPARISON WITH PROMPT-TUNING

Prompt tuning is a popular alternative for PEFT where learnable tokens are appended to human-designed prompts and optimized on to improve accuracy. We choose to report the Maple Khattak et al. (2023a) + DePT Zhang et al. (2024b) state-of-the-art configuration as it is shown in Zhang et al. (2024b) to be a highly competitive configuration for image classification. Table 5 reports the results for 4 and 16 shots over the 11 datasets used in Zhang et al. (2024b). We train on ViT-B/32 with all algorithms training approximately 3M parameters. We report that although competitive for low shots, prompt tuning struggles to keep up in the 16-shot setting. We note in particular that prompt tuning struggles on datasets that require more adaptation (e.g. FGVCAircraft) whereas LoRA and RandLoRA in particular manage to more largely improve results. We additionally report that Maple + DePT requires a much longer training time and VRAM usage. For example, 16-shots on ImageNet requires 3.5h and 18GB of VRAM for Maple + DePT while it requires 2 minutes and 4.5GB of VRAM for RandLoRA. Because prompt tuning is largely orthogonal to LoRA-type weight updates we suggest that future research should study how to combine these approaches together.

Table 5: Comparison of LoRA and RandLoRA with a state-of-the-art prompt tuning algorithm. CLIP ViT-B/32.

| Shots | Method | ImageNet | Caltech101 | OxfordIIITPet | Cars | Flowers102 | Food101 | FGVCAircraft | SUN397 | DTD | EuroSAT | UCF101 | Average |
|---|---|---|---|---|---|---|---|---|---|---|---|---|---|
| 4 | LoRA-16 | **64.9** | 92.0 | 88.2 | 63.9 | 87.9 | **82.6** | 30.3 | 68.2 | 61.1 | **89.4** | 74.7 | 73.0 |
| | RandLoRA-10 | 63.9 | 91.7 | 86.4 | 67.0 | 89.9 | 80.8 | **34.0** | 69.7 | **62.4** | 84.4 | 74.9 | 73.2 |
| | Maple + DePT | 62.1 | **95.0** | **89.5** | **68.7** | **90.5** | 79.6 | 28.3 | **70.2** | 61.7 | 81.4 | **76.6** | 73.1 |
| 16 | LoRA-16 | 65.8 | 91.7 | 89.5 | **80.1** | 94.9 | 81.8 | 42.5 | 73.5 | 72.0 | 91.2 | 81.5 | 78.6 |
| | RandLoRA-10 | 66.3 | 95.6 | **91.1** | 77.4 | 94.5 | **84.0** | **45.0** | 73.7 | **72.5** | **94.1** | 81.7 | **79.6** |
| | Maple + DePT | **67.7** | **96.0** | 90.5 | 79.1 | **96.3** | 81.7 | 36.9 | **74.5** | 70.3 | 90.3 | **82.1** | 78.7 |

### B.4 COMMONSENSE REASONING RESULTS FOR DORA

We compare RandLoRA with DoRA (Liu et al., 2024) for tuning LLama3 in Table 6. We find that RandLoRA outperforms both DoRA and LoRA for larger parameter budgets (rank 32), while DoRA and LoRA are competitive at "Efficient" budgets (rank 16).

## C IMPLEMENTATION DETAILS

### C.1 CLASSIFICATION DATASETS

We fine-tune vision architectures on 22 vision datasets (21 for pure vision backbones where ImageNet is removed for brevity). We train for 10 epochs on the few-shot experiments and increase the

Table 6: Further comparison with DoRA related methods on LLama3-8b. Results averaged over 8 commonsense reasoning tasks. We bold the best accuracy.

| Method | Efficient | | Performant | |
|---|---|---|---|---|
| | 15k | 170k | 15k | 170k |
| LoRA | 82.7 | 84.4 | **83.1** | 85.2 |
| DoRA | **82.8** | 84.3 | 82.5 | 85.2 |
| RandLoRA | 81.0 | **84.6** | 81.3 | **85.6** |

number of epochs according to dataset constraints for $50\%$ and $100\%$ fine-tuning. Table 7 reports details of the 22 datasets we use as well as the number of epochs used as in (Zhang et al., 2024a).

Table 7: Vision datasets used for the image classification experiments

| # | Datasets | Classes | Splits | | | Epochs |
|---|---|---|---|---|---|---|
| | | | train | val | test | |
| (1) | Cars | 196 | 7,330 | 814 | 8,041 | 35 |
| (2) | DTD | 47 | 3,384 | 376 | 1,880 | 76 |
| (3) | EuroSAT | 10 | 21,600 | 2,700 | 2,700 | 12 |
| (4) | GTSRB | 43 | 23,976 | 2,664 | 12,630 | 11 |
| (5) | MNIST | 10 | 55,000 | 5,000 | 10,000 | 5 |
| (6) | RESISC45 | 45 | 17,010 | 1,890 | 6,300 | 15 |
| (7) | SUN397 | 397 | 17,865 | 1,985 | 19,850 | 14 |
| (8) | SVHN | 10 | 68,257 | 5,000 | 26,032 | 4 |
| (9) | CIFAR10 | 10 | 45,000 | 5,000 | 10,000 | 5 |
| (10) | CIFAR100 | 100 | 45,000 | 5,000 | 10,000 | 6 |
| (11) | ImageNet | 1,000 | 1,276,167 | 5,000 | 50,000 | 10 |
| (12) | STL10 | 10 | 4,500 | 500 | 8,000 | 4 |
| (13) | Food101 | 101 | 70,750 | 5,000 | 25,250 | 15 |
| (14) | Caltech101 | 101 | 6,941 | 694 | 1,736 | 10 |
| (15) | Caltech256 | 257 | 22,037 | 2,448 | 6,122 | 8 |
| (16) | FGVCAircraft | 100 | 3,334 | 3,333 | 3,333 | 60 |
| (17) | Flowers102 | 102 | 1,020 | 1,020 | 6,149 | 40 |
| (18) | OxfordIIITPet | 37 | 3,312 | 368 | 3,669 | 5 |
| (19) | CUB200 | 200 | 5,395 | 599 | 5,794 | 20 |
| (20) | PascalVOC | 20 | 7,844 | 7,818 | 14,976 | 10 |
| (21) | Country211 | 211 | 31,650 | 10,550 | 21,100 | 15 |
| (22) | UCF101 | 101 | 7,639 | 1,898 | 3,783 | 20 |

## C.2 CLIP

We utilize the pytorch AdamW optimizer with weight decay $0.1$ and a cosine decaying learning rate schedule. To accommodate the full batch size on a single A100 GPU for the ViT-L/14 and ViT-H/14 CLIP architectures, we accumulate 2 batches of 64. This is excepted for the standard fine-tuning of the ViT-H/14 for standard fine-tuning where we need to accumulate 4 batches of 32 due to increasing memory costs. We acquire the pre-trained weights from the openclip repository (Cherti et al., 2023) where the use the "openai" weights from ViT-B/32 and ViT-L/14 and the "laion2b_s32b_b79k" weights for ViT-H/14.

## C.3 PURE VISION BACKBONES

For pure vision backbones, we use the same configuration as vision and language fine-tuning of CLIP except that we increase the learning rate to $10^{-2}$ for LoRA and RandLoRA. We train RandLoRA-6 for ViT-B/32 and RandLoRA-8 for Dinov2's ViT-B/14 and CLIP's ViT-L/14.

## C.4 COMMONSENSE REASONING

Our evaluation protocol assesses the model's versatility and reasoning capabilities across eight diverse datasets: BoolQ (Clark et al., 2019) (yes/no question answering), PIQA (Bisk et al., 2020) (physics commonsense questions), SIQA (Sap et al., 2019) (social implications reasoning), HellaSwag (Zellers et al., 2019) (multi-choice scenario completion), WinoGrande (Sakaguchi et al.,

Table 8: Hyper-parameters for different algorithms. Multiple values for hyperparameters denote variances accross the ViT-B/32, ViT-L/14 and ViT-H/14 architectures respectively.

| Algorithm | FT | LoRA | NoLA | VeRA | RandLoRA |
|---|---|---|---|---|---|
| Batch size | 128/64/32 | | | 128/64/64 | |
| Learning Rate (LR) | 1e-5 | 1e-3 | 1e-3 | 1e-2 | 1e-3 |
| Scaling coefficient | 1 | $\frac{1}{r}$ | $\frac{1}{r}$ | $\frac{1}{r}$ | $\frac{10}{r}$ |
| Basis rank (r) | – | 32 | 1 | 256/256/1024 | 6/8/10 |
| Number of basis ($n$) | – | – | 1024 | 1 | 128 |

Table 9: LLM fine-tuning hyper-parameters for different algorithms. Multiple values for hyper-parameters denote variances accross the Qwen2 -0.5b, Phi3-8b and LLama3-8b architectures respectively.

| Algorithm | LoRA | NoLA | VeRA | RandLoRA |
|---|---|---|---|---|
| Batch size | | | 16/8/4 | |
| Learning Rate (LR) | | | $10^{-4}$ | |
| Scaling coefficient | 2 | $\frac{2}{\sqrt{n}}$ | 2 | $\frac{2}{\sqrt{n}}$ |
| Basis rank (r) | 32 | 1 | 256/1024/1024 | 6/10/15 |
| Number of basis ($n$) | – | 1024 | 1 | 149/153/136 |

2021) (binary sentence completion), ARC-c and ARC-e (Clark et al., 2018) (challenging and easy science questions at a grade-school level), and OBQA (Mihaylov et al., 2018) (multi-step reasoning). These datasets collectively pose a wide range of challenges, from natural language understanding and commonsense reasoning to physical and social inference. For further details on these datasets, we refer readers to the survey by Hu et al. (Hu et al., 2023). We train using the hugginface[2] transformers library and follow the implementation[3] of Liu et al (Liu et al., 2024). We train for 3 epochs using a learning rate of $1 \times 10^{-4}$ and a base scaling coefficient of 2 for the weight update. To prevent overfitting, we add a dropout layer in each of the adapter's layers with a dropout probability of $0.05$ and perform early stopping using the same validation set of size 120, drawn from the training set. We maintain hyper-parameters the same across architectures and algorithms except for the scaling ratio of the weight update for NoLA and RandLoRA which we further multiply by $1/\sqrt{n}$ where $n$ is the number of bases to account for the increasing norm of the sum of random matrices.

## C.5 TRAINING TIME, MEMORY CONSUMPTION AND RANDOM BASES

### C.5.1 REDUCING MEMORY CONSUMPTION

**Basis sharing across layers** RandLoRA aims to preserve the memory efficiency and training speed advantages of LoRA. As shown in Section 4, although RandLoRA trains an amount of parameters comparable to LoRA we still have to store $N$ large random bases for each weight update. We first note that as observed in previous research, (Koohpayegani et al., 2024; Kopiczko et al., 2024) random bases can be shared across layers. In practice, we generate one pair of random matrices $B_i \in \mathbb{R}^{N \times D_m \times r}$ and $A_0 \in \mathbb{R}^{\times r \times d_m}$, where $D_m$ and $d_m$ represent the largest $D$ and $d$ across all network layers. During forward and backward passes on a layer of size $D \times d$, we select the first $D$ rows of $B$ and $d$ columns of $A$ to perform the weight update. This strategy stores only the largest $B$ and $A$ matrices, which would have to be fit in memory at some point during training anyways. Note that although we do not study this case, this strategy directly generalizes to having different ranks $r$ across layers as has been proposed in AutoLoRA (Zhang et al., 2024c) for example. This strategy allows us to avoid increasing memory as network depth increases, meaning that RandLoRA become more efficient when network depth increases.

---

[2]https://huggingface.co
[3]https://github.com/NVlabs/DoRA/tree/main/commonsense_reasoning

**Efficient back-propagation with a single random $A$ basis**  We evidence in section 4.2 that the $A_i$ matrices do not need to be $N$ dimensional and that a single $A$ matrix modified by $N$ $\Gamma_i$ is enough to acheive full rank. We can thus optimize the backward pass when computing the gradient of $\Lambda_i$ and $Gamma_i$ so that we only have to store one matrix $A \in \mathbb{R}^{r \times d}$ for the backward pass, further reducing memory consumption.

**Efficient matrix multiplication in the forward pass**  We adopt the notations from Section C.5.1 to optimize the matrix multiplication of $X \in \mathbb{R}^{B \times D}$ during the forward and backward passes: $XW$. Given the pre-trained weight $W_0 \in \mathbb{D} \times$, LoRA computes $Y = XW_0 + XBA$ where we compute $Y = XW_0 + \sum_{i=1}^{N} (XB_i)(\Lambda_i A_i \Gamma_i)$. These equations suggest RandLoRA would be $N$ times slower to run than LoRA but in practice, the $XW_0$ operation dominates the matmul time and the $N$ RandLoRA operations are naturally parallelized by the CUDA kernel. In practice we observe a $13\%$ training time increase for the smaller ViT-B-32 models and up to $100\%$ in the worst case for larger models with large weight matrices such as LLama3.

## C.6 Sparse random bases

We continue here the discussion on the possible collinearity of sparse bases. We remind here that we construct the random bases $B_i$ and $A_i$ by assigning

$$\begin{cases} -1, & \text{with probability } \frac{1}{s} \\ 0, & \text{with probability } 1 - \frac{2}{s} \\ 1, & \text{with probability } \frac{1}{s} \end{cases}$$

where $s$ an integer in $[2, \sqrt{D}]$ for $W \in \mathbb{R}^{D \times d}$. Because of the ternary nature of these matrices, there is a non-zero probability that two row are collinear across all random matrices, resulting in non full rank. If we can show that is probability is negligible then the full rank constraint will be preserved in practice. We compute that the probability of drawing the same size $d$ row twice equates to $p = 2 \times (\frac{s^2 - 4s + 6}{s^2})^d$. Taking the example of the ViT-B/32 architectures with $W \in \mathbb{R}^{768 \times 768}$ and for the largest recommended optimal sparsity ($s = \sqrt{768}$) we compute $p = 2 \times 10^{-49}$. The probability of drawing at least two collinear row over $N$ matrices of is $p_2 = (N + D)(N + D - 1)p$. In the RandLoRA-6 configuration for ViT-B, $N = 128$ resulting in $p_2 = 8 \times 10^{-44}$ meaning these events are negligible in practice even with a large number of sparse bases and that the full rank constraint is preserved.

### C.6.1 Training time

We report in Table 10 the relative training time of RandLoRA compared to LoRA and standard fine-tuning on a single RTX4090 GPU (A100 for LLama3 and ViT-H/14). Since we do not have ressources to fully fine-tune LLama3, we report LoRA as the memory baseline. In addition to Table 10 we report up to $212\%$ increase over LoRA-64 training time for the best performing RandLoRA-15 configuration for LLama3-8b. This number should be put in perspective with DoRA leading to a $220\%$ increase in all configurations for LLama3-8b.

## D Mathematical derivations and proofs

### D.1 Theorem 4.1

In this section we would like to give the details of the proof of theorem 4.1 from the main paper. In order to do so we will start by proving a few lemmas.

Our method consider decompositions similar to those given in equation 1 and equation 2 that are built from random matrices instead of the left and right singular vectors. A key observation is that such decompositions and their sums will yield high rank matrix approximations. The following two lemmas explains why this is the case.

| Model | Architecture | LoRA-32 | DoRA-32 | RandLoRA | FT |
|---|---|---|---|---|---|
| CLIP-ViT-B/32 | Training Time | 90 | – | 113 | 100% |
| | Memory | 81 | – | 78 | 100% |
| CLIP-ViT-L/14 | Training Time | 95 | – | 128 | 100% |
| | Memory | 72 | – | 71 | 100% |
| CLIP-ViT-H/14 | Training Time | 96 | – | 122 | 100% |
| | Memory | 54 | – | 51 | 100% |
| LLama3-8B | Training Time | 100 | 220 | 167 | – |
| | Memory | 100 | 102 | 102 | – |

Table 10: Comparison of training times for LoRA, RandLoRA, and FT on vision-language or language architectures.

**Lemma D.1.** *Let $B = [B_1, \ldots, B_n]$ denote a matrix where each $B_j \in \mathbb{R}^{D \times r}$ and let $A = [A_1, \ldots, A_n]$ denote a matrix where each $A_j \in \mathbb{R}^{d \times r}$. Assume $nr \leq \min(D, d)$ and assume that the columns of $B$ are linearly independent and the columns of $A$ are linearly independent. Define*

$$C = \sum_{j=1}^{n} B_j A_j^{\mathsf{T}} \tag{9}$$

*Then we must have that $rank(C) = nr$.*

*Proof.* We first observe that using the inequality $rank(X + Y) \leq rank(X) + rank(Y)$ we get that $rank(C) \leq nr$ because each term $B_j A_j^{\mathsf{T}}$ has rank $r$, since the columns of $A$ and $B$ are linearly independent, and there are $n$ of them.

Then observe that we can rewrite $C$ as

$$C = BA^T \tag{10}$$

Using Sylvester's rank inequality: If $X \in \mathbb{R}^{D \times l}$ and $Y \in \mathbb{R}^{l \times d}$ then

$$rank(X) + rank(Y) - l \leq rank(XY) \tag{11}$$

we have that

$$rank(C) = rank(BA^T) \tag{12}$$
$$\geq rank(B) + rank(A^T) - kj \tag{13}$$
$$= 2nr - nr \tag{14}$$
$$= nr \tag{15}$$

and the proof is complete. $\square$

**Lemma D.2.** *Let $\{X_1, \ldots, X_n\}$ denote $n$ vectors in $\mathbb{R}^N$ where $n \leq N$ drawn i.i.d from a Gaussian or uniform distribution. Then with probability $1$ $\{X_1, \ldots, X_n\}$ will be linearly independent.*

*Proof.* We first note that any measure defined via a Gaussian or Uniform probability distribution is absolutely continuous with respect to the Lebesgue measure. Meaning they have the same sets of measure zero as the Lebesgue measure.

We then prove the case that $\{X_1, \ldots, X_n\}$ are vectors of unit length. Since the vectors were drawn independently, we can first assume we drew $X_1$. The probability that this is the zero vector is $0$ w.r.t the Lebesgue measure on the closed unit ball $B_N(0)$ about the origin in $\mathbb{R}^N$ and hence any other measure absolutely continuous to it. Then draw $X_2$ and note that the probability that $X_2$ lies in $span\{X_1\} \cap B_N(0)$ is also $0$ since $span\{X_1\} \cap B_N(0)$ forms a set of $0$ Lebesgue measure in $B_N(0)$. Continuing in this way we find that $\{X_1, \ldots, X_n\}$ will be linearly independent with probability $1$.

For the general case where $\{X_1, \ldots, X_n\}$ are not drawn to have unit length i.e. drawn on the sphere in $\mathbb{R}^N$, we simply note that we can draw each one and then divide by its norm producing one of unit length. Since normalizing by the norm doesn't affect linear independence we get by the above case that $\{X_1, \ldots, X_n\}$ must be linearly independent with probability $1$. $\square$

Lemmas D.1 and D.2 show that if we were to i.i.d draw $n$ random vectors $A_1, \ldots, A_n$ in $\mathbb{R}^D$ and $n$ vectors $B_1, \ldots, B_n$ using a Gaussian or uniform distribution for $n \leq \min(D, d)$. Then the matrix $Q = AB^T$ would have rank $n$, where $A = [A_1, \ldots, A_n]$ and $B = [B_1, \ldots, B_n]$.

We note that lemma D.1 is still true if we were to consider products of the form $B\Lambda A\Gamma$, where $\Lambda$ and $\Gamma$ are diagonal matrices with non-zero diagonal entries.

Using the above two lemmas we can now give a proof of theorem 4.1 from the main paper.

*Proof.* The fact that each $B_i\Lambda_i A_i\Gamma_i$ has rank $r$ with probability 1 follows from lemmas D.1 and D.2. In order to estimate the difference $\|W - \sum_{j=1}^n B_j\Lambda_j A_j\Gamma_j\|$, we use equation 2 to write

$$W = \sum_{j=1}^n U_j\Sigma_j V_j^\mathsf{T}. \tag{16}$$

We can then estimate

$$\|W - \sum_{j=1}^n B_j\Lambda_j A_j\Gamma_j\|_F = \|\sum_{j=1}^n U_j\Sigma_j V_j^\mathsf{T} - \sum_{j=1}^n B_j\Lambda_j A_j\Gamma_j\|_F \tag{17}$$

$$= \|\sum_{j=1}^n U_j\Sigma_j V_j^\mathsf{T} - B_j\Lambda_j A_j\Gamma_j\|_F \tag{18}$$

$$\leq \sum_{j=1}^n \|U_j\Sigma_j V_j^\mathsf{T} - B_j\Lambda_j A_j\Gamma_j\|_F \tag{19}$$

$$\leq n \cdot \epsilon \tag{20}$$

where the last inequality follows from the assumption equation 6. $\qquad\square$

### D.2 LoRA's low bound

We demonstrate here the short derivation leading to the results of equation equation 8.

*Proof.* By definition, the forbenius norm of a matrix $X \in \mathbb{R}^{n \times n}$, $\|X\|_F$ is invariant under left and right multiplications by any orthogonal matrices $P \in \mathbb{R}^{n \times n}$ and $Q \in \mathbb{R}^{n \times n}$, i.e. $\|X\|_F = \|PXQ\|_F$. Then, given the k-truncated SVD of $M = U\Sigma_k V^T$ with $U, V \in \mathbb{R}^{n \times n}$ and $\Sigma_k \in \mathbb{R}^{n \times n}$ diagonal with elements above the $k$-th being 0, $U$ and $V$ are orthogonal matrices by definition. We then have the following,

$$\|X - M\|_F = \|U(X - M)V^T\|_F \tag{21}$$

$$= \|\Sigma - \Sigma_k\|_F \tag{22}$$

$$= \sum_{j=k+1}^r \sigma_j^2 \tag{23}$$

where $\Sigma \in \mathbb{R}^{n \times n}$ is diagonal and contains the $n$ singular values of $X$ by decreasing order and $\sigma_j$ denotes the $j$-th element of $\Sigma$.

Since by the SVD definition, the best rank-$k$ approximation of $W$ is $M$, given LoRA's rank-$k$ approximation of $W$ by the matrix multiplication $BA$ where $B \in \mathbb{R}^{n \times k}$ and $A \in \mathbb{R}^{k \times n}$ we have

$$\|X - M\|_F \leq \|X - BA\|_F \tag{24}$$

$$\sum_{j=k+1}^r \sigma_j^2 \leq \|X - BA\|_F. \tag{25}$$

$\qquad\square$

Table 11: Detailed accuracy results per dataset, fine-tuning the vision and language backbones of CLIP-ViT-B/32. Highest performance and those within a range of 0.1 in each section are highlighted in bold.

| | Method | Cars | DTD | EuroSAT | GTSRB | MNIST | RESISC45 | SUN397 | SVHN | CIFAR10 | CIFAR100 | ImageNet | STL10 | Food101 | Caltech256 | FGVCAircraft | Flowers102 | OxfordIIITPet | CUB200 | PascalVOC | Country211 | Caltech101 | UCF101 | Average |
|---|---|---|---|---|---|---|---|---|---|---|---|---|---|---|---|---|---|---|---|---|---|---|---|---|
| **1 shot** | NoLA | 51.6 | 44.5 | 72.8 | 54.3 | 76.3 | 64.1 | 53.8 | 31.1 | 81.3 | 62.7 | 49.7 | 90.4 | 61.9 | 76.6 | 19.0 | 62.5 | 69.7 | 41.8 | 69.1 | 5.3 | 84.9 | 61.4 | 58.4 |
| | VeRA256 | **60.9** | 47.7 | **76.8** | 47.4 | 71.7 | 67.4 | **64.9** | **47.5** | **90.4** | **71.7** | **63.7** | **97.4** | **83.5** | **83.3** | **22.1** | 68.5 | **88.3** | **54.4** | **77.6** | **17.6** | 87.5 | 64.9 | **66.1** |
| | LoRA32 | 51.9 | 46.3 | 73.2 | 61.4 | 73.7 | 67.9 | 53.9 | 30.6 | 79.8 | 63.9 | 51.7 | 89.5 | 63.5 | 78.1 | 19.1 | 65.3 | 69.9 | 43.0 | 67.1 | 5.6 | 85.2 | 63.5 | 59.3 |
| | RandLoRA6 | 53.6 | **50.3** | 73.1 | 61.4 | **78.5** | **72.6** | 59.3 | 29.4 | 80.8 | 67.1 | 57.4 | 92.6 | 69.8 | 81.5 | 21.7 | **71.3** | 75.0 | 48.5 | 67.6 | 8.5 | **88.3** | **67.0** | 62.5 |
| | FT | 51.4 | 46.8 | 67.3 | **62.8** | 77.4 | 69.9 | 57.2 | 20.0 | 68.3 | 61.1 | 52.2 | 83.0 | 66.7 | 79.5 | 19.0 | 68.7 | 70.0 | 46.5 | 59.0 | 7.4 | 86.1 | 66.6 | 58.5 |
| **2 shots** | NoLA | 57.1 | 54.3 | 82.8 | 63.6 | 83.2 | 69.7 | 57.9 | 32.2 | 80.3 | 68.5 | 51.0 | 92.2 | 67.2 | 80.4 | 24.3 | 72.7 | 80.8 | 47.3 | 57.9 | 7.4 | 85.2 | 65.4 | 62.8 |
| | VeRA256 | **62.1** | 49.5 | 71.0 | 50.5 | 72.2 | 68.1 | **64.8** | **50.7** | **91.7** | **73.1** | **63.7** | **97.5** | **84.2** | **84.0** | 22.1 | 69.9 | **89.2** | 54.8 | **73.8** | **17.7** | **89.2** | 65.0 | 66.6 |
| | LoRA32 | 53.7 | 56.9 | 82.0 | 62.6 | 82.8 | 71.9 | 60.1 | 36.8 | 84.2 | 71.5 | 52.9 | 94.1 | 73.6 | 82.8 | 22.4 | 73.8 | 84.2 | 48.0 | 61.7 | 9.0 | 87.8 | 67.4 | 64.6 |
| | RandLoRA6 | 59.5 | **60.4** | **83.4** | 73.7 | **85.2** | **74.9** | 62.0 | 30.0 | 82.6 | 72.0 | 57.7 | 94.5 | 72.0 | 83.8 | **28.6** | 80.8 | 83.7 | 54.3 | 62.3 | 9.8 | 89.0 | **71.7** | **66.9** |
| | FT | 58.5 | 57.7 | 82.9 | **76.7** | 84.8 | 74.4 | 60.3 | 23.0 | 69.4 | 68.3 | 53.9 | 87.3 | 69.1 | 83.0 | 26.2 | **81.0** | 79.1 | **55.2** | 53.2 | 9.3 | **89.2** | 71.6 | 64.3 |
| **4 shots** | NoLA | 60.1 | 58.1 | 86.9 | 67.7 | 87.5 | 75.0 | 61.0 | 45.3 | 87.2 | 69.3 | 51.4 | 91.3 | 72.3 | 81.2 | 26.0 | 80.7 | 84.1 | 51.6 | 69.0 | 9.3 | 87.3 | 68.0 | 66.8 |
| | VeRA256 | 61.8 | 49.6 | 79.7 | 52.5 | 73.2 | 69.6 | **64.9** | **52.2** | **92.3** | **73.9** | **64.2** | **97.5** | **84.9** | 83.8 | 21.9 | 70.4 | **89.5** | 54.9 | **75.8** | **17.8** | 89.4 | 65.6 | 67.5 |
| | LoRA32 | 57.0 | 60.4 | 86.7 | 59.0 | 86.5 | 73.5 | 62.3 | 46.4 | 87.1 | 71.1 | 52.5 | 93.6 | 76.3 | 83.2 | 24.2 | 77.2 | 84.7 | 50.9 | 69.5 | 11.2 | 88.4 | 67.1 | 66.8 |
| | RandLoRA6 | 63.1 | **63.2** | 87.9 | 77.4 | **88.2** | **80.3** | 65.0 | 47.8 | 87.6 | 72.9 | 55.8 | 93.2 | 74.8 | **84.1** | 31.1 | 87.8 | 85.0 | 58.8 | 70.3 | 10.7 | **89.8** | 75.3 | **70.4** |
| | FT | **65.2** | 60.3 | 85.4 | **82.5** | 87.0 | 80.1 | 64.1 | 41.1 | 78.9 | 70.8 | 54.0 | 84.3 | 72.0 | 83.2 | **34.1** | **89.5** | 80.1 | **60.1** | 62.5 | 10.0 | 89.6 | 73.8 | 68.6 |
| **16 shots** | NoLA | 66.2 | 66.5 | 92.3 | 73.6 | 91.2 | 81.2 | 64.4 | **74.9** | 92.1 | 74.3 | 54.0 | 95.0 | 77.3 | 84.0 | 30.4 | 86.0 | 89.6 | 61.1 | 73.5 | 12.0 | 88.2 | 73.7 | 72.8 |
| | VeRA256 | 62.9 | 51.4 | 82.4 | 53.2 | 75.8 | 70.5 | 66.3 | 57.0 | **93.3** | 73.9 | **64.6** | **97.9** | **85.2** | 85.6 | 22.3 | 71.6 | **90.9** | 55.8 | **76.4** | **18.1** | 89.2 | 65.7 | 68.6 |
| | LoRA32 | 69.6 | 64.8 | 87.5 | 61.2 | 91.2 | 79.8 | 65.0 | 71.6 | 93.0 | 75.7 | 54.9 | 95.8 | 77.3 | 85.8 | 33.7 | 83.3 | 89.6 | 64.4 | 75.2 | 12.1 | 88.5 | 76.3 | 72.6 |
| | RandLoRA6 | 71.9 | **70.2** | **94.2** | 81.5 | 94.1 | 84.9 | **67.6** | 73.7 | 92.0 | **77.0** | 56.8 | 95.0 | 80.1 | **86.9** | 35.1 | 91.3 | 89.3 | 68.6 | 75.5 | 12.2 | **90.9** | **79.3** | 75.8 |
| | FT | **74.0** | 69.8 | 93.2 | **87.5** | **94.3** | 86.7 | 67.2 | 74.1 | 89.8 | 76.3 | 56.2 | 92.7 | 78.6 | **86.9** | **39.1** | **93.2** | 89.0 | **70.1** | 74.9 | 12.1 | **90.9** | 78.9 | **76.2** |
| **50%** | NoLA | 69.7 | 68.9 | 98.6 | 93.9 | 98.7 | 91.6 | 64.9 | 93.0 | 97.1 | 79.0 | 56.9 | 97.8 | 81.0 | 86.3 | 44.2 | 81.9 | 89.6 | 62.3 | 85.6 | 14.4 | 88.9 | 78.0 | 78.3 |
| | VeRA256 | 63.7 | 62.4 | 95.5 | 79.2 | 92.8 | 81.1 | 66.3 | 75.6 | 95.2 | 76.3 | **64.6** | **97.9** | 85.6 | 87.9 | 25.6 | 72.1 | 88.8 | 56.6 | 85.4 | **18.1** | 93.3 | 70.7 | 74.3 |
| | LoRA32 | 71.9 | 71.3 | 98.4 | 94.7 | 98.8 | 93.0 | 65.6 | 93.7 | 97.4 | 81.5 | 59.5 | 97.7 | 85.4 | 88.1 | 45.3 | 85.8 | 89.2 | 65.2 | 86.5 | 14.1 | 88.5 | 80.2 | 79.6 |
| | RandLoRA6 | **78.0** | 73.6 | 98.5 | 95.5 | 99.0 | 94.0 | 67.4 | 94.6 | **97.7** | 84.4 | 62.4 | 97.9 | **87.6** | **89.5** | 56.3 | 88.5 | 90.0 | **70.3** | 86.5 | 14.6 | **95.3** | **82.5** | **82.0** |
| | FT | **78.0** | 72.4 | **98.7** | **96.2** | **99.1** | **94.5** | 67.0 | **95.0** | 97.6 | **84.8** | 62.1 | **98.0** | 86.6 | 89.2 | **57.4** | **89.1** | **91.1** | 69.0 | **87.2** | 14.6 | 94.9 | 81.8 | **82.0** |
| **100%** | NoLA | 73.6 | 73.5 | 98.8 | 95.2 | 99.0 | 93.3 | 66.4 | 94.2 | 97.6 | 80.3 | 57.5 | 98.1 | 82.0 | 87.5 | 51.1 | 89.1 | 90.8 | 67.1 | 86.5 | 15.9 | 90.1 | 78.4 | 80.3 |
| | VeRA256 | 63.7 | 62.5 | 95.2 | 79.5 | 92.2 | 80.6 | 66.3 | 75.4 | 95.2 | 76.2 | **64.6** | 98.1 | 85.6 | 87.8 | 25.4 | 77.3 | 90.6 | 56.8 | 85.9 | **18.1** | 93.8 | 70.3 | 74.6 |
| | LoRA32 | 77.3 | 76.7 | 98.6 | 95.3 | 99.1 | 94.4 | 67.1 | 95.2 | 97.9 | 83.8 | 60.5 | **98.4** | 87.8 | 89.2 | 59.5 | 91.4 | 91.1 | 70.7 | 87.7 | 15.9 | 89.6 | 82.0 | 82.2 |
| | RandLoRA6 | 83.1 | **78.9** | **99.0** | 96.1 | **99.3** | 95.4 | 69.5 | 95.5 | **98.1** | **87.0** | 63.8 | **98.4** | **89.4** | **90.9** | 67.1 | 93.7 | 91.0 | **75.2** | **88.0** | 16.8 | 95.6 | **85.1** | 84.4 |
| | FT | **84.4** | 77.7 | 98.9 | **96.8** | 99.2 | **96.0** | 69.0 | **96.0** | 97.9 | 86.9 | 63.7 | **98.5** | 88.8 | 90.8 | **68.1** | **94.8** | 91.2 | 74.8 | **88.0** | 16.3 | **95.8** | 84.6 | **84.5** |

# E DETAILED RESULTS

## E.1 VISION LANGUAGE: CLIP

We report per dataset accuracies for NoLA, VeRA, LoRA, standard fine-tuning (FT) and RandLoRA in for the CLIP ViT-B/32 ViT-L/14 and ViT-H/14 architectures on 22 datasets in Tables 11, 12and 13 respectively.

## E.2 VISION ONLY: DINOV2

Table 14 reports detailed results when fine-tuning DinoV2 on 21 datasets. We use the pre-trained ViT-B/14 architecture and train a linear classifier together with the feature extractor. Compared to the CLIP results ImageNet was removed to promote brevity of the experiments.

## E.3 COMMONSENSE REASONING

Table 15 reports detailed accuracy results for the Qwen2, Phi3 and LLama3 language models trained on the commonsense tasks. See C.4 for details on the datasets and the hyper-parameters used.

Table 12: Detailed accuracy results per dataset, fine-tuning the vision and language backbones of CLIP-ViT-L/14. Highest performance and those within a range of 0.1 in each section are highlighted in bold.

| | Method | Cars | DTD | EuroSAT | GTSRB | MNIST | RESISC45 | SUN397 | SVHN | CIFAR10 | CIFAR100 | ImageNet | STL10 | Food101 | Caltech256 | FGVCAircraft | Flowers102 | OxfordIIITPet | CUB200 | PascalVOC | Country211 | Caltech101 | UCF101 | Average |
|---|---|---|---|---|---|---|---|---|---|---|---|---|---|---|---|---|---|---|---|---|---|---|---|---|
| 1 shot | NoLA | 72.7 | 61.1 | 81.5 | 76.4 | 89.3 | 78.6 | 67.3 | **76.1** | 94.0 | 77.9 | 70.3 | 98.8 | 87.5 | 88.3 | 41.1 | 85.0 | 90.4 | 63.4 | 71.7 | 18.0 | 90.4 | 76.6 | 75.3 |
| | VeRA256 | 78.5 | 55.6 | 75.3 | 55.0 | 88.8 | 73.2 | 68.8 | 67.8 | **96.6** | 80.5 | 75.5 | **99.4** | **93.2** | 88.9 | 34.3 | 80.6 | **93.8** | 64.2 | **78.8** | **32.0** | 86.8 | 73.6 | 74.6 |
| | LoRA32 | 74.9 | 62.3 | 81.0 | 76.5 | 91.7 | 79.5 | 68.3 | 74.7 | 92.8 | 78.9 | 71.4 | 98.6 | 87.9 | 89.4 | **44.0** | 89.5 | 88.7 | 66.3 | 68.5 | 19.3 | 90.3 | 77.7 | 76.0 |
| | RandLoRA10 | 76.8 | **63.1** | **83.5** | 72.5 | **92.7** | 81.6 | **74.7** | 74.2 | 95.0 | **83.0** | **76.2** | 99.3 | 91.6 | **92.1** | 43.2 | 89.1 | 91.0 | 68.8 | 74.9 | 27.2 | 90.3 | **82.7** | **78.3** |
| | FT | 73.6 | 62.4 | 81.2 | **78.4** | **92.8** | **83.8** | 71.5 | 68.3 | 91.0 | 81.3 | 73.2 | 98.6 | 88.4 | 91.6 | 41.8 | **90.5** | 88.7 | **68.9** | 66.1 | 23.0 | **90.7** | 82.5 | 76.7 |
| 2 shots | NoLA | 74.0 | 66.7 | 81.1 | 81.2 | 93.2 | 82.4 | 68.0 | 78.3 | 93.3 | 80.8 | 66.3 | 98.2 | 88.0 | 89.4 | 39.6 | 92.5 | 93.9 | 64.8 | 75.2 | 20.5 | 91.2 | 76.6 | 77.1 |
| | VeRA256 | 78.1 | 55.8 | 75.3 | 55.7 | 90.0 | 73.5 | 68.6 | 67.0 | **96.6** | 81.3 | **75.6** | **99.4** | **93.2** | 89.0 | 34.8 | 81.5 | **94.4** | 64.0 | **79.2** | **32.2** | 86.8 | 74.1 | 74.8 |
| | LoRA32 | 77.3 | 68.1 | 84.7 | 82.7 | **95.2** | 84.2 | 69.9 | **78.5** | 92.4 | 81.6 | 68.7 | 97.8 | 88.7 | 90.0 | 46.4 | 94.5 | 91.8 | 69.3 | 72.6 | 21.0 | 91.7 | 79.5 | 78.5 |
| | RandLoRA10 | 78.5 | 70.4 | **85.1** | 80.4 | 94.7 | 85.8 | **74.9** | 78.2 | 95.9 | **84.1** | 74.4 | **99.5** | 91.9 | **92.5** | 46.1 | 94.5 | 93.9 | 71.5 | 75.8 | 28.1 | 91.7 | 83.6 | **80.5** |
| | FT | **79.6** | 70.5 | 83.8 | **84.0** | 94.0 | 86.5 | 73.2 | 78.1 | 92.5 | 82.7 | 72.1 | 99.2 | 89.2 | 91.6 | **47.2** | **96.5** | 93.1 | **73.5** | 72.7 | 24.1 | **91.9** | **84.1** | 80.0 |
| 4 shots | NoLA | 75.2 | 70.0 | 87.4 | 85.5 | 95.5 | 84.4 | 69.2 | 82.5 | 94.8 | 82.2 | 66.2 | 97.9 | 89.3 | 89.7 | 44.5 | 93.1 | 94.2 | 67.3 | 77.0 | 23.0 | 91.3 | 77.2 | 79.0 |
| | VeRA256 | 77.9 | 56.7 | 77.8 | 56.0 | 91.3 | 74.1 | 69.8 | 68.0 | 96.9 | 81.4 | **75.9** | **99.5** | **93.2** | 89.1 | 35.1 | 81.1 | 94.6 | 64.2 | **79.3** | **32.1** | 86.9 | 74.2 | 75.2 |
| | LoRA32 | 77.2 | 71.8 | 88.4 | 86.2 | 95.9 | 86.3 | 70.5 | **84.3** | 95.1 | 82.4 | 68.7 | 97.5 | 90.2 | 90.8 | 47.4 | 95.5 | 93.7 | 70.6 | 75.8 | 23.2 | 91.8 | 81.4 | 80.2 |
| | RandLoRA10 | 79.3 | 73.6 | 89.2 | 85.2 | **96.4** | 87.8 | **74.6** | 80.9 | **97.3** | 85.1 | 72.6 | 99.3 | 92.4 | 92.4 | 47.1 | 93.7 | **94.8** | 71.0 | 79.1 | 29.2 | 91.7 | 84.6 | **81.7** |
| | FT | **79.7** | **74.6** | **90.0** | **90.1** | 96.0 | **88.8** | 73.5 | 82.5 | 94.2 | 84.2 | 71.6 | 98.1 | 89.8 | **92.7** | 43.3 | **97.3** | 93.7 | **76.0** | 78.2 | 25.3 | **92.3** | **84.8** | 81.7 |
| 16 shots | NoLA | 82.8 | 72.0 | 93.7 | 86.4 | 96.7 | 87.3 | 72.2 | 87.8 | 97.0 | 84.2 | 69.1 | 98.7 | 90.5 | 93.0 | 53.5 | 96.2 | 94.6 | 78.8 | **83.7** | 23.6 | 90.3 | 82.7 | 82.5 |
| | VeRA256 | 80.5 | 56.1 | 82.6 | 56.2 | 93.9 | 74.4 | 71.9 | 69.8 | **97.2** | 83.0 | **76.3** | **99.5** | **93.5** | 90.3 | 38.3 | 82.3 | 94.8 | 68.3 | 80.2 | **32.8** | 89.1 | 77.2 | 76.7 |
| | LoRA32 | 85.7 | 74.8 | 94.2 | 88.1 | 97.1 | 88.9 | 73.3 | **88.7** | 96.9 | 85.8 | 70.9 | 99.0 | 91.2 | 93.2 | 56.7 | 97.5 | 94.2 | 82.6 | 82.1 | 23.6 | 90.8 | 85.5 | 83.7 |
| | RandLoRA10 | 86.6 | 76.0 | 94.9 | 87.4 | 97.2 | 89.4 | **76.5** | 86.4 | 97.0 | 86.5 | 74.5 | 99.2 | 92.3 | **94.4** | 57.4 | 97.8 | **95.3** | 83.9 | 82.4 | 25.3 | 91.7 | **88.5** | 84.6 |
| | FT | **87.5** | **78.4** | **95.7** | **91.7** | **97.7** | **91.2** | 75.6 | 87.4 | 94.6 | **87.3** | 73.5 | 98.3 | 91.4 | 94.1 | **61.1** | **98.4** | 94.2 | **85.0** | 82.3 | 25.8 | **92.9** | 88.2 | **85.1** |
| 50% | NoLA | 84.4 | 78.0 | 98.6 | 96.4 | **99.3** | 95.2 | 72.7 | 96.3 | 99.1 | 89.2 | 73.0 | **99.5** | 93.0 | 94.4 | 57.9 | 96.3 | 95.6 | 79.3 | 91.5 | 26.7 | 91.3 | 87.0 | 86.1 |
| | VeRA256 | 81.7 | 68.8 | 95.8 | 88.5 | 97.0 | 86.8 | 71.8 | 90.5 | 98.1 | 85.0 | 76.2 | **99.5** | 93.9 | 93.8 | 44.5 | 87.7 | 94.4 | 70.2 | 88.6 | **32.9** | 94.3 | 80.9 | 82.8 |
| | LoRA32 | 88.2 | 81.2 | 98.8 | **96.9** | 99.1 | 96.0 | 74.1 | 96.5 | **99.2** | 90.3 | 75.4 | **99.5** | 94.4 | 95.6 | 68.4 | **97.2** | 94.9 | 83.2 | 91.0 | 25.5 | 94.1 | 88.4 | 87.6 |
| | RandLoRA10 | **89.9** | **82.3** | 98.8 | 96.8 | **99.4** | 96.0 | **76.7** | 96.8 | **99.2** | 91.6 | **78.3** | 99.5 | 94.7 | 95.6 | 69.0 | 96.9 | **95.7** | 83.9 | **92.1** | 27.5 | **96.9** | 90.5 | **88.5** |
| | FT | 89.7 | 79.0 | **99.1** | 96.3 | 99.3 | **96.8** | 76.0 | **97.0** | **99.2** | 91.2 | 77.3 | 99.4 | 94.3 | 95.8 | **69.6** | 97.1 | 95.0 | **84.6** | 91.9 | 26.8 | **96.9** | 90.8 | 88.3 |
| 100% | NoLA | 87.5 | 82.5 | 99.0 | 96.8 | 99.3 | 96.3 | 75.0 | 96.6 | 99.3 | 90.4 | 73.6 | **99.7** | 93.8 | 95.2 | 74.0 | 98.5 | 95.1 | 83.2 | 91.5 | 28.5 | 93.9 | 88.0 | 88.1 |
| | VeRA256 | 81.6 | 67.9 | 96.1 | 88.6 | 97.2 | 85.8 | 71.7 | 90.2 | 98.2 | 85.1 | 77.0 | 99.3 | 93.8 | 93.9 | 44.5 | 93.0 | 94.9 | 70.3 | 89.2 | **32.8** | 96.4 | 81.5 | 83.1 |
| | LoRA32 | 89.2 | 83.9 | **99.2** | 97.4 | 99.3 | 96.8 | 75.9 | 95.8 | **99.3** | 91.4 | 76.1 | **99.7** | 95.2 | 95.8 | 78.6 | 98.4 | 95.2 | 85.3 | 91.7 | 27.9 | 96.1 | 90.2 | 89.0 |
| | RandLoRA10 | **90.8** | **84.6** | 99.0 | 96.6 | **99.5** | 96.9 | **77.8** | 97.0 | **99.4** | 92.8 | 79.0 | 99.7 | 95.4 | 96.5 | 79.6 | 98.9 | **95.4** | 87.1 | 92.5 | 30.4 | 96.8 | **93.1** | **90.0** |
| | FT | 90.4 | 84.4 | **99.1** | 97.1 | 99.3 | **97.2** | 77.2 | **97.3** | 99.2 | 92.4 | 78.1 | **99.6** | 94.9 | 96.2 | **81.5** | **99.1** | 94.8 | 86.9 | **92.6** | 29.3 | **97.0** | 92.6 | 89.8 |

Table 13: Detailed accuracy results per dataset, fine-tuning the vision and language backbones of CLIP-ViT-H/14. Highest performance and those within a range of 0.1 in each section are highlighted in bold.

| | Method | Cars | DTD | EuroSAT | GTSRB | MNIST | RESISC45 | SUN397 | SVHN | CIFAR10 | CIFAR100 | ImageNet | STL10 | Food101 | Caltech256 | FGVCAircraft | Flowers102 | OxfordIIITPet | CUB200 | PascalVOC | Country211 | Caltech101 | UCF101 | Average |
|---|---|---|---|---|---|---|---|---|---|---|---|---|---|---|---|---|---|---|---|---|---|---|---|---|
| 1 shot | NoLA | 92.0 | **71.8** | 80.9 | 78.7 | **90.1** | 82.2 | 70.6 | **60.2** | 95.1 | 82.7 | 73.1 | 98.0 | 88.0 | 90.3 | 46.4 | 91.5 | 91.2 | 76.2 | 69.7 | 17.5 | 91.5 | 78.1 | 78.0 |
| | VeRA1024 | **93.8** | 69.4 | 73.8 | 65.1 | 90.0 | 73.2 | 74.9 | 54.2 | **98.2** | 85.5 | 77.6 | **99.1** | **92.8** | 91.5 | 46.4 | 81.6 | 92.0 | **82.2** | **80.0** | **29.9** | 89.7 | 79.2 | 78.2 |
| | LoRA32 | 92.7 | 70.4 | **84.6** | **79.8** | 88.2 | 84.7 | 71.2 | 59.9 | 95.6 | 83.3 | 71.9 | 96.7 | 87.5 | 90.6 | 49.0 | 95.2 | 90.4 | 76.6 | 70.2 | 18.4 | 91.8 | 79.8 | 78.6 |
| | RandLoRA10 | 93.0 | 71.0 | 79.8 | 79.6 | **90.2** | 84.3 | **78.3** | 55.8 | 97.2 | **85.9** | **78.0** | 98.1 | 90.9 | **92.5** | **49.9** | 94.3 | **92.2** | 78.3 | 66.1 | 26.4 | **92.3** | 82.1 | **79.8** |
| | FT | 92.2 | 69.9 | 81.7 | **79.8** | 88.2 | **85.3** | 76.2 | 56.2 | 95.8 | 83.3 | 73.3 | 97.6 | 89.1 | 91.7 | **49.8** | **95.6** | 90.6 | 76.9 | 66.1 | 24.6 | 91.9 | **82.4** | 79.0 |
| 2 shots | NoLA | 92.8 | 71.7 | 89.3 | 87.8 | 91.2 | 83.2 | 71.8 | 75.1 | 96.0 | 84.3 | 68.9 | 95.5 | 88.6 | 91.0 | 50.7 | 94.6 | 92.3 | 79.4 | 71.7 | 20.8 | 91.6 | 78.8 | 80.3 |
| | VeRA1024 | **93.8** | 71.1 | 89.7 | 67.0 | 90.3 | 74.3 | **78.2** | 74.3 | **98.1** | 85.8 | **77.3** | **99.0** | **92.9** | 91.7 | 47.0 | 82.1 | 92.3 | **81.7** | **80.8** | **30.1** | 89.5 | 79.4 | 80.3 |
| | LoRA32 | 93.1 | 71.9 | **93.2** | 84.9 | 92.2 | 86.0 | 72.3 | 77.4 | 96.9 | 83.5 | 70.4 | 94.6 | 87.7 | 91.9 | 51.9 | 97.2 | 91.8 | 77.5 | 75.5 | 20.8 | 92.9 | 83.6 | 81.2 |
| | RandLoRA10 | **93.9** | **75.8** | 90.7 | 89.3 | 93.5 | **86.9** | **78.2** | **79.0** | 97.5 | **86.5** | 74.8 | 98.1 | 91.3 | **92.5** | **53.6** | **97.4** | **93.2** | 81.0 | 72.6 | 27.2 | **93.7** | 84.1 | **83.2** |
| | FT | 93.1 | 74.0 | 90.8 | **89.9** | **93.7** | 86.1 | 74.3 | 74.0 | 95.4 | 85.2 | 71.2 | 97.2 | 90.1 | 92.0 | 46.9 | **97.4** | 92.6 | 78.2 | 71.5 | 25.1 | 93.3 | **84.5** | 81.7 |
| 4 shots | NoLA | 93.1 | 73.7 | 92.9 | 86.3 | 94.4 | 85.1 | 72.6 | 80.3 | 96.8 | 84.2 | 69.2 | 97.2 | 89.4 | 91.0 | 55.2 | 95.8 | 92.7 | 80.7 | 78.7 | 23.5 | 91.6 | 82.4 | 82.1 |
| | VeRA1024 | 93.9 | 71.4 | 92.4 | 67.3 | 92.5 | 74.2 | **76.6** | 78.0 | **98.3** | 86.1 | 72.9 | **99.1** | **92.8** | 92.8 | 47.6 | 82.0 | **94.0** | 82.7 | **82.2** | **30.8** | 89.9 | 79.6 | 80.8 |
| | LoRA32 | 93.9 | 73.7 | 94.2 | **89.5** | 95.6 | 87.8 | 72.5 | **80.9** | 97.1 | 85.3 | 70.8 | 97.3 | 89.1 | 92.3 | 56.9 | 97.8 | 92.4 | 82.4 | 78.5 | 23.3 | 91.9 | 84.8 | 83.1 |
| | RandLoRA10 | **94.1** | **78.6** | **95.5** | **89.5** | 95.7 | **89.8** | 76.5 | 80.5 | 98.1 | **87.4** | **73.5** | 99.0 | 91.6 | 92.7 | **57.5** | 98.1 | 93.7 | **83.1** | 78.1 | 28.6 | **93.3** | **86.9** | **84.6** |
| | FT | 93.8 | 78.1 | 94.0 | 88.5 | **95.8** | 89.4 | 75.5 | 77.1 | 96.2 | 86.5 | 72.8 | 97.7 | 90.5 | **93.5** | 51.8 | **98.4** | 92.8 | 81.5 | 77.2 | 25.2 | 93.0 | 86.3 | 83.4 |
| 16 shots | NoLA | 93.3 | 76.0 | 95.7 | 90.3 | 96.7 | 88.6 | 75.3 | 87.1 | **98.0** | 87.3 | 71.6 | 98.7 | 90.0 | 92.8 | 61.5 | 98.1 | 93.7 | 86.0 | 81.5 | 23.8 | 92.2 | 85.9 | 84.7 |
| | VeRA1024 | 94.2 | 77.4 | 94.3 | 81.7 | 94.4 | 85.1 | 77.1 | 82.0 | **98.4** | 87.8 | 73.8 | **99.3** | **91.7** | 94.0 | 61.1 | 94.6 | **94.5** | 86.4 | 81.2 | **25.7** | 93.2 | 88.1 | 84.4 |
| | LoRA32 | 93.5 | 77.7 | 95.3 | 92.5 | 96.6 | 90.2 | 75.7 | 86.8 | 98.2 | 88.3 | 73.2 | 98.5 | 90.4 | 93.9 | **65.5** | 98.8 | 92.9 | 86.8 | 80.9 | 23.2 | 92.2 | 87.7 | 85.4 |
| | RandLoRA10 | **94.4** | 79.8 | **95.9** | 92.3 | **96.9** | **91.7** | **78.1** | **87.4** | 98.1 | **88.6** | **75.6** | 99.0 | 91.2 | **94.6** | 64.5 | **99.0** | 94.0 | **87.8** | 80.7 | 25.2 | 92.2 | **89.1** | **86.2** |
| | FT | 94.3 | **80.0** | **95.8** | 94.1 | 96.5 | **91.7** | 77.5 | 85.6 | 97.8 | 87.6 | 75.0 | 98.6 | 90.8 | 94.5 | 64.7 | 98.7 | 93.7 | 87.1 | **81.9** | 25.2 | **93.3** | 89.6 | 86.1 |
| 50% | NoLA | 93.0 | 80.8 | **99.1** | 96.7 | **99.2** | 95.2 | 75.2 | 96.2 | 99.2 | 90.7 | 74.7 | 99.3 | 93.0 | 95.3 | 70.5 | 97.5 | 94.5 | 85.2 | 90.9 | 25.9 | 91.7 | 87.4 | 87.8 |
| | VeRA1024 | 93.8 | 82.0 | **99.2** | 96.2 | **99.3** | 96.1 | 76.9 | 96.2 | 99.2 | 91.9 | 76.3 | **99.5** | 93.6 | 96.2 | 72.7 | 98.3 | **95.2** | 86.7 | 90.5 | 25.7 | 95.8 | 89.4 | 88.7 |
| | LoRA32 | 92.7 | 82.1 | 98.8 | 96.5 | **99.3** | 96.2 | 75.7 | 96.5 | **99.3** | 91.9 | 77.0 | 99.4 | 94.2 | 96.0 | 74.0 | 97.2 | 94.5 | 86.4 | 89.5 | 25.9 | **96.2** | 90.5 | 88.6 |
| | RandLoRA10 | **94.8** | **82.8** | 98.8 | 96.6 | **99.3** | 96.4 | **77.7** | **96.8** | **99.3** | **93.0** | **79.1** | **99.5** | 94.6 | **96.5** | 77.2 | **98.7** | 94.7 | **87.3** | **91.3** | **28.1** | 96.0 | 90.4 | **89.5** |
| | FT | 94.5 | 82.0 | 98.8 | **96.8** | **99.3** | **96.6** | 76.7 | **96.8** | 99.0 | 91.8 | 77.0 | 99.4 | 94.2 | **96.4** | **77.9** | 98.1 | 94.6 | 86.7 | 90.8 | 27.6 | 95.5 | **91.4** | 89.2 |
| 100% | NoLA | 93.3 | 84.2 | **99.3** | 96.7 | 99.4 | 96.2 | 76.6 | 96.8 | 99.2 | 91.5 | 74.8 | 99.3 | 93.7 | 95.5 | 77.2 | 98.7 | 94.4 | 86.9 | 91.4 | 28.2 | 93.0 | 89.8 | **89.0** |
| | VeRA1024 | 94.3 | 85.0 | 99.0 | 97.2 | 99.4 | 97.0 | 78.0 | 96.8 | **99.3** | 92.6 | 76.7 | **99.6** | 94.3 | 95.9 | 79.7 | **99.3** | 95.0 | 87.7 | 91.3 | 27.3 | 96.4 | 91.3 | 89.7 |
| | LoRA32 | 93.1 | 85.8 | 99.1 | 97.3 | **99.5** | 97.2 | 77.6 | 97.2 | **99.3** | 93.0 | 77.9 | **99.5** | 94.9 | 96.6 | 83.7 | 99.0 | 94.4 | 87.5 | **91.8** | 28.3 | 95.9 | 91.5 | 90.0 |
| | RandLoRA10 | 94.7 | **86.0** | 99.0 | 97.0 | **99.4** | 97.1 | **79.4** | **97.3** | **99.3** | **93.5** | **80.1** | **99.5** | **95.2** | **97.1** | 84.1 | **99.3** | 95.1 | **88.6** | 91.7 | **31.2** | **96.5** | 92.7 | **90.6** |
| | FT | **94.9** | 84.2 | 98.8 | **97.5** | **99.5** | 97.6 | 78.7 | **97.3** | 99.2 | 92.8 | 77.9 | **99.5** | 94.8 | 96.8 | **84.4** | **99.3** | 95.3 | 88.3 | **91.8** | 30.1 | **96.6** | **93.0** | 90.4 |

Table 14: Detailed accuracy results per dataset, the DinoV2 ViT-B/14 vision backbone. Highest performance and those within a range of 0.1 in each section are highlighted in bold.

| | Method | Cars | DTD | EuroSAT | GTSRB | MNIST | RESISC45 | SUN397 | SVHN | CIFAR10 | CIFAR100 | STL10 | Food101 | Caltech256 | FGVCAircraft | Flowers102 | OxfordIIITPet | CUB200 | PascalVOC | Country211 | Caltech101 | UCF101 | Average |
|---|---|---|---|---|---|---|---|---|---|---|---|---|---|---|---|---|---|---|---|---|---|---|---|
| 1 shots | NoLA | 21.2 | 45.4 | 60.7 | 28.8 | 55.0 | **49.8** | 46.3 | 14.4 | 73.8 | **57.3** | 71.7 | 50.5 | **78.7** | 19.7 | 98.6 | 74.6 | 62.5 | **43.6** | 3.1 | 85.5 | **63.8** | 52.6 |
| | VeRA256 | 22.5 | 45.6 | 57.9 | 20.1 | 50.7 | 44.6 | **46.7** | 12.9 | 76.5 | 55.8 | 64.4 | 51.9 | 78.6 | 19.1 | 98.7 | 75.5 | 62.9 | 36.2 | **3.4** | 84.7 | 63.1 | 51.0 |
| | LoRA32 | **22.6** | 47.2 | 59.3 | 24.8 | 51.7 | 48.7 | 45.9 | 14.6 | **77.2** | **57.4** | 64.6 | **52.4** | 77.5 | **19.7** | **98.9** | **76.5** | **63.1** | 37.2 | **3.4** | 85.1 | 62.3 | 51.9 |
| | RandLoRA6 | 21.5 | **47.8** | 57.9 | **34.5** | **61.6** | 44.9 | 44.1 | **16.2** | 56.8 | 54.0 | 66.0 | 47.2 | 76.4 | **19.6** | 97.8 | 71.8 | 59.9 | 43.4 | 3.0 | **86.1** | 62.8 | 51.1 |
| | FT | 20.8 | 45.5 | **67.8** | 25.7 | 52.3 | 45.2 | 45.3 | 15.5 | 70.2 | 54.7 | **75.8** | 50.1 | 75.2 | 19.4 | 98.3 | 70.8 | 60.0 | 36.9 | 3.1 | 84.7 | 60.3 | 51.3 |
| 2 shots | NoLA | 41.4 | 57.8 | 64.1 | 43.1 | **73.5** | **65.4** | 58.1 | 16.2 | **90.0** | 74.5 | **93.9** | 64.5 | 84.9 | 28.0 | **99.6** | 83.0 | 73.3 | 51.4 | 4.1 | **90.0** | 72.7 | **63.3** |
| | VeRA256 | 38.2 | 57.4 | 64.0 | 28.9 | 65.0 | 60.8 | 57.3 | 14.4 | 86.0 | 71.6 | 78.9 | **66.2** | 84.8 | 26.2 | 99.4 | 83.4 | **74.8** | 44.4 | 4.1 | 88.0 | **73.7** | 60.4 |
| | LoRA32 | 41.1 | 58.4 | 68.3 | 37.7 | 71.4 | 64.7 | **58.2** | 14.7 | 89.6 | **74.8** | 87.0 | 66.1 | **85.3** | 27.0 | 99.5 | 86.7 | 73.0 | **52.1** | 5.0 | 89.2 | 72.1 | 63.0 |
| | RandLoRA6 | 41.9 | **59.6** | **69.0** | **48.6** | 70.2 | 62.2 | 57.1 | **19.4** | 72.1 | 70.6 | 84.8 | 63.3 | 83.8 | **29.5** | 98.4 | 80.0 | 71.3 | 49.7 | 3.8 | **89.9** | 72.1 | 61.8 |
| | FT | **43.1** | 56.0 | 65.7 | 42.6 | 72.1 | 63.1 | 57.5 | 16.1 | 79.0 | 71.0 | 91.3 | 64.6 | 85.1 | 27.7 | 99.3 | 81.1 | 71.8 | 50.6 | 3.9 | 89.2 | 70.5 | 62.0 |
| 4 shots | NoLA | 62.9 | **68.5** | 76.4 | 62.4 | 82.4 | 75.9 | 65.8 | 22.8 | **94.5** | **82.6** | **97.6** | 73.7 | 89.8 | 40.6 | **99.7** | 89.3 | 82.6 | 65.4 | 6.0 | 90.6 | 79.5 | **71.9** |
| | VeRA256 | 56.1 | 64.2 | 71.5 | 43.2 | 76.1 | 71.6 | 64.8 | 17.6 | 91.4 | 80.9 | 88.7 | 74.7 | 89.3 | 36.0 | **99.7** | 91.0 | 82.1 | 53.5 | 6.2 | 89.8 | 77.7 | 67.9 |
| | LoRA32 | 63.4 | 66.5 | **79.2** | 61.0 | 79.2 | **77.6** | **66.3** | 20.9 | **94.5** | 82.1 | 94.5 | **75.0** | 89.4 | 41.9 | **99.7** | **91.9** | **83.5** | **66.2** | 6.8 | 89.7 | **80.6** | **71.9** |
| | RandLoRA6 | 64.6 | 65.3 | 72.2 | **66.6** | **86.4** | 77.0 | 65.0 | **24.8** | 84.0 | 79.4 | 93.1 | 73.0 | 89.8 | **43.9** | 99.6 | 86.6 | 82.4 | 63.8 | 5.9 | **91.7** | 78.6 | 71.1 |
| | FT | **65.5** | 67.3 | 73.0 | 62.7 | 85.6 | 73.8 | 66.0 | 20.9 | 88.0 | 81.3 | 94.0 | 73.6 | **90.2** | 41.4 | 99.6 | 88.9 | 82.6 | 61.3 | 6.0 | **91.7** | 79.3 | 71.1 |
| 16 shots | NoLA | 86.4 | 79.5 | 91.0 | 88.2 | 94.0 | 87.8 | **75.1** | 56.9 | 97.2 | **89.4** | 99.0 | 85.1 | 93.1 | 64.0 | **99.7** | 93.9 | **88.8** | **82.2** | 12.2 | **95.3** | 87.0 | 83.1 |
| | VeRA256 | 81.7 | 78.2 | 88.2 | 60.1 | 88.9 | 83.2 | 73.5 | 30.2 | **97.4** | 87.6 | 97.4 | 84.4 | 92.6 | 51.3 | **99.7** | **94.6** | 88.5 | 72.9 | 11.8 | 91.9 | 85.7 | 78.1 |
| | LoRA32 | 87.1 | **80.5** | **93.9** | 86.4 | 93.0 | 87.2 | **75.1** | 44.8 | **97.4** | 88.8 | **99.4** | **85.6** | **93.5** | 65.2 | **99.7** | 94.2 | 88.5 | 80.5 | **12.4** | 94.6 | 87.6 | 82.6 |
| | RandLoRA6 | **88.4** | 79.0 | 92.3 | **90.3** | **95.4** | 87.3 | 74.7 | **57.4** | 97.0 | 88.5 | 98.2 | 85.5 | 93.1 | **71.5** | **99.7** | 93.3 | 88.6 | 79.7 | 11.8 | 94.5 | **87.8** | **83.5** |
| | FT | 87.3 | 78.8 | 92.4 | 88.9 | 95.0 | **88.9** | 74.7 | 50.4 | 96.8 | 88.6 | 98.8 | 85.3 | 93.4 | 67.1 | **99.7** | 93.2 | **88.9** | 77.8 | 11.8 | 94.9 | 87.4 | 82.9 |
| 0.5 shots | NoLA | 89.0 | 82.5 | 98.9 | 96.4 | 99.2 | 94.8 | 76.0 | 96.5 | **99.2** | 93.2 | **99.6** | 92.5 | **95.8** | 73.7 | 99.5 | 94.9 | 87.4 | 92.8 | 18.7 | **97.8** | 88.6 | 88.9 |
| | VeRA256 | 84.0 | 80.1 | 97.3 | 89.7 | 97.7 | 92.0 | 74.8 | 88.2 | 99.0 | 92.2 | 99.4 | 91.7 | 94.7 | 68.4 | 99.5 | **95.1** | 86.9 | 89.9 | 17.6 | 96.0 | 87.5 | 86.7 |
| | LoRA32 | 89.7 | 82.8 | **99.0** | 96.2 | 99.1 | 94.8 | 75.9 | 96.6 | **99.3** | **93.7** | 99.5 | 93.2 | 95.5 | 72.6 | 98.3 | 95.0 | 88.6 | **93.0** | 19.0 | 97.5 | 90.3 | 89.0 |
| | RandLoRA6 | 89.7 | **83.2** | 98.7 | **97.1** | **99.3** | 95.5 | 75.8 | **97.2** | **99.2** | 93.4 | **99.6** | 93.3 | 95.5 | 75.2 | **99.7** | 94.9 | 87.5 | 92.8 | 19.6 | 97.6 | 89.3 | **89.2** |
| | FT | **90.3** | 81.5 | 98.8 | 96.6 | **99.3** | 95.8 | 76.2 | 96.6 | **99.2** | 93.4 | 99.3 | 93.0 | 95.7 | **75.3** | 98.9 | 95.0 | 87.4 | 92.4 | **19.9** | 97.3 | **90.6** | **89.2** |
| 1.0 shots | NoLA | 92.5 | 85.4 | 98.8 | 96.9 | **99.3** | 96.1 | 77.8 | 96.8 | **99.4** | 94.1 | **99.7** | 93.4 | **96.2** | 81.8 | **99.7** | 95.9 | 90.2 | **93.6** | 22.3 | **98.2** | 90.2 | 90.4 |
| | VeRA256 | 89.8 | 81.6 | 97.4 | 89.5 | 98.1 | 93.1 | 76.5 | 88.4 | 99.1 | 92.6 | **99.6** | 92.5 | 95.3 | 75.4 | **99.7** | 95.8 | 89.7 | 90.6 | 20.5 | 97.1 | 88.2 | 88.1 |
| | LoRA32 | 92.7 | 84.6 | 99.1 | 96.3 | **99.3** | 96.0 | 78.2 | 97.2 | 99.3 | 94.2 | **99.7** | 93.7 | **96.3** | 83.3 | **99.7** | 95.7 | **90.4** | 92.7 | 20.6 | 97.8 | 91.5 | 90.4 |
| | RandLoRA6 | **93.3** | **85.5** | 99.0 | **97.1** | 99.4 | 96.8 | 77.9 | **97.5** | **99.5** | 94.4 | **99.7** | 94.2 | **96.2** | 84.0 | **99.6** | 95.8 | 90.1 | 93.1 | 22.7 | 98.0 | **92.0** | 90.8 |
| | FT | **93.4** | **85.5** | **99.4** | 96.8 | **99.3** | 97.0 | 78.4 | 97.4 | 99.2 | 94.0 | **99.6** | 94.1 | **96.3** | 83.9 | **99.7** | 95.8 | 90.1 | 93.1 | **23.8** | 98.0 | 91.8 | 90.8 |

| Method | % Params | BoolQ | PIQA | SIQA | HellaSwag | WinoGrande | ARC-e | ARC-c | OBQA | Average + Δ |
|---|---|---|---|---|---|---|---|---|---|---|
| | | | | | Qwen2 - Zero-shot | | | | | |
| Zero-shot | 0 | 3.12 | 4.68 | 7.22 | 2.50 | 14.52 | 4.80 | 1.79 | 2.60 | 5.15 |
| | | | | | Qwen2 - 15k | | | | | |
| NoLA | 0.05 | 54.16 | 56.91 | 47.65 | 17.36 | 45.46 | 46.55 | 32.51 | 39.80 | 42.55 |
| VeRA1024 | 0.06 | 58.78 | 56.64 | 50.10 | 24.95 | 49.80 | 56.52 | 37.80 | 50.40 | 48.12 |
| LoRA-16 | 1.18 | 62.14 | 62.13 | 58.24 | 27.86 | 49.96 | 62.46 | 44.97 | 58.20 | 53.25 |
| RandLoRA-10 | 1.18 | 62.14 | 63.49 | 55.32 | 31.16 | 49.96 | 64.27 | 44.97 | 56.60 | 53.49 +0.24 |
| LoRA-32 | 2.33 | 59.94 | 62.13 | 56.55 | 30.27 | 41.99 | 64.39 | 46.42 | 57.00 | 52.34 |
| RandLoRA-5 | 2.33 | 62.81 | 63.82 | 54.86 | 30.00 | 48.07 | 64.81 | 43.34 | 55.40 | 52.89 +0.55 |
| | | | | | Qwen2 - 170k | | | | | |
| NoLA | 0.05 | 55.99 | 52.50 | 55.07 | 23.74 | 50.51 | 55.64 | 38.91 | 46.80 | 47.40 |
| VeRA1024 | 0.06 | 55.50 | 59.30 | 52.81 | 34.52 | 52.72 | 58.55 | 42.94 | 57.80 | 51.78 |
| LoRA-16 | 1.18 | 53.39 | 68.12 | 66.33 | 46.46 | 58.72 | 59.97 | 43.77 | 62.20 | 57.37 |
| RandLoRA-10 | 1.18 | 61.47 | 67.63 | 65.61 | 40.26 | 57.22 | 62.12 | 47.95 | 59.60 | 57.73 +0.36 |
| LoRA-32 | 2.33 | 55.78 | 68.28 | 67.20 | 42.37 | 60.22 | 61.03 | 45.05 | 58.80 | 57.34 |
| RandLoRA-5 | 2.33 | 63.46 | 65.72 | 66.43 | 42.90 | 56.20 | 61.49 | 47.53 | 59.20 | 57.86 +0.52 |
| | | | | | Phi3 - Zero-shot | | | | | |
| Zero-shot | 0 | 62.26 | 79.82 | 65.81 | 56.29 | 19.89 | 89.86 | 77.65 | 71.40 | 65.37 |
| | | | | | Phi3 - 15k | | | | | |
| NoLA | 0.005 | 66.24 | 85.15 | 73.49 | 78.29 | 73.95 | 95.33 | 85.15 | 85.20 | 80.35 |
| VeRA1024 | 0.015 | 68.53 | 84.49 | 73.08 | 74.54 | 72.85 | 93.01 | 80.97 | 81.60 | 78.63 |
| LoRA-16 | 0.57 | 69.51 | 85.36 | 75.44 | 80.15 | 75.85 | 95.37 | 86.09 | 86.60 | 81.80 |
| RandLoRA-40 | 0.58 | 69.54 | 85.31 | 73.80 | 84.05 | 75.14 | 94.65 | 84.90 | 85.80 | 81.65 -0.15 |
| LoRA-32 | 1.14 | 68.44 | 85.31 | 74.67 | 72.14 | 74.98 | 95.20 | 85.41 | 86.60 | 80.34 |
| RandLoRA-20 | 1.16 | 69.20 | 85.42 | 75.33 | 83.98 | 75.77 | 95.50 | 85.92 | 87.60 | 82.33 +1.99 |
| LoRA-64 | 2.28 | 69.88 | 85.75 | 74.97 | 74.45 | 75.30 | 95.54 | 87.12 | 88.00 | 81.37 |
| RandLoRA-10 | 2.29 | 69.63 | 85.31 | 75.03 | 86.94 | 75.30 | 95.24 | 85.58 | 86.40 | 82.43 +1.06 |
| | | | | | Phi3 - 170k | | | | | |
| NoLA | 0.005 | 68.87 | 85.15 | 77.18 | 85.13 | 77.90 | 95.20 | 85.58 | 83.60 | 82.33 |
| VeRA1024 | 0.015 | 69.53 | 84.53 | 74.52 | 84.08 | 76.82 | 94.51 | 83.68 | 83.54 | 81.40 |
| LoRA-16 | 0.57 | 70.83 | 84.39 | 78.45 | 89.94 | 82.87 | 95.45 | 86.09 | 89.00 | 84.63 |
| RandLoRA-40 | 0.58 | 70.86 | 86.67 | 78.81 | 90.07 | 82.00 | 95.12 | 86.26 | 87.60 | 84.67 +0.04 |
| LoRA-32 | 1.14 | 71.23 | 85.96 | 78.92 | 91.77 | 82.95 | 94.61 | 84.81 | 89.40 | 84.96 |
| RandLoRA-20 | 1.16 | 71.62 | 87.43 | 79.48 | 91.48 | 82.79 | 95.16 | 86.01 | 87.80 | 85.22 +0.26 |
| LoRA-64 | 2.28 | 71.93 | 86.13 | 79.58 | 90.14 | 83.74 | 92.68 | 81.74 | 87.80 | 84.22 |
| RandLoRA-10 | 2.29 | 71.87 | 86.56 | 79.43 | 90.99 | 82.72 | 95.66 | 85.49 | 87.40 | 85.01 +0.79 |
| | | | | | LLama3 - Zero-shot | | | | | |
| Zero-shot | 0 | 60.73 | 41.40 | 28.40 | 25.00 | 10.97 | 16.41 | 15.96 | 16.80 | 26.96 |
| | | | | | LLama3 - 15k | | | | | |
| NoLA | 0.004 | 67.58 | 84.49 | 72.31 | 69.60 | 70.56 | 90.49 | 78.75 | 81.20 | 76.87 |
| VeRA1024 | 0.014 | 63.36 | 84.39 | 74.10 | 77.70 | 71.35 | 89.48 | 76.54 | 80.20 | 77.14 |
| LoRA-16 | 0.35 | 73.03 | 86.94 | 75.90 | 90.53 | 77.74 | 90.74 | 80.29 | 86.20 | 82.67 |
| RandLoRA-60 | 0.36 | 71.19 | 84.22 | 75.59 | 83.82 | 74.98 | 91.12 | 81.31 | 86.00 | 81.03 -1.64 |
| LoRA-32 | 0.7 | 74.22 | 86.40 | 75.79 | 91.90 | 77.35 | 90.61 | 80.80 | 87.60 | 83.09 |
| RandLoRA-30 | 0.7 | 71.65 | 83.79 | 74.56 | 86.85 | 75.61 | 90.78 | 80.03 | 87.20 | 81.31 -1.78 |
| LoRA-64 | 1.4 | 71.77 | 84.17 | 76.25 | 85.14 | 73.80 | 91.46 | 80.80 | 86.20 | 81.20 |
| RandLoRA-15 | 1.4 | 70.98 | 86.02 | 75.44 | 89.74 | 76.80 | 91.29 | 81.66 | 83.80 | 81.96 +0.76 |
| | | | | | LLama3 - 170k | | | | | |
| NoLA | 0.004 | 71.83 | 84.66 | 77.79 | 85.05 | 82.72 | 88.59 | 76.45 | 82.20 | 81.16 |
| VeRA1024 | 0.014 | 70.55 | 85.69 | 79.27 | 92.14 | 82.64 | 87.33 | 73.38 | 82.20 | 81.65 |
| LoRA-16 | 0.35 | 75.14 | 89.12 | 80.66 | 89.01 | 86.58 | 90.07 | 78.75 | 86.20 | 84.44 |
| RandLoRA-60 | 0.35 | 75.26 | 87.98 | 79.63 | 94.66 | 85.64 | 90.03 | 79.44 | 84.40 | 84.62 +0.18 |
| LoRA-32 | 0.7 | 75.08 | 88.85 | 80.25 | 95.42 | 86.19 | 90.28 | 80.29 | 85.60 | 85.24 |
| RandLoRA-30 | 0.7 | 76.33 | 88.08 | 80.25 | 95.67 | 86.11 | 90.36 | 80.89 | 87.00 | 85.59 +0.45 |
| LoRA-64 | 1.4 | 74.65 | 89.66 | 80.86 | 95.17 | 86.74 | 90.95 | 79.18 | 85.40 | 85.33 |
| RandLoRA-15 | 1.4 | 72.63 | 87.98 | 81.37 | 95.68 | 87.77 | 91.33 | 80.89 | 89.00 | 85.83 +0.50 |

Table 15: Comparison of accuracy on commonsense reasoning datasets. We report accuracy delta of RandLoRA with LoRA for comparable amounts of trainable parameters.