# OpenReview forum: "RandLoRA: Full rank parameter-efficient fine-tuning of large models"
_ICLR.cc/2025/Conference — ICLR 2025 Poster_

### Official Review · Reviewer_ZGK2 · 2024-10-31

**Soundness:** 3
**Presentation:** 2
**Contribution:** 2
**Rating:** 6
**Confidence:** 4

**Summary:**

This paper introduces RandLoRA, a novel method for parameter-efficient fine-tuning (PEFT) of large pre-trained models. By leveraging learned linear combinations of low-rank, non-trainable random matrices, RandLoRA enables full-rank updates, which significantly enhance the adaptability and efficiency of fine-tuning processes. The method strategically limits the number of trainable parameters by optimizing diagonal scaling matrices, which are applied to the fixed random bases, thus maintaining a low parameter count and minimal memory usage during training.

**Strengths:**

(1): The manuscript is well-crafted with a clear and logical progression of ideas.
(2): Visual aids like figures and tables are effectively used to illustrate key points and compare performance metrics clearly.
(3): The extensive experiments across various tasks and architectures demonstrate the method's effectiveness and adaptability.

**Weaknesses:**

(1): Lines 86-89: The phenomenon of performance saturation as the rank of LoRA increases is well-known in the field (This has already been explained in the VeRA paper.). I suggest that this point be rephrased or discussed within the context of known literature to maintain the integrity of the paper.
(2): While the method is promising in terms of parameter efficiency and memory usage, its practicality is challenged by the substantially increased training times on the Llama3B model. A more thorough investigation into the computational trade-offs and possible optimizations to reduce training times would benefit the study and its broader applicability.

**Questions:**

(1): Lines 77-80: The paper claims that RandLoRA consistently outperforms LoRA across the same parameter counts. However, based on Figure 1(a) and 1(b), it appears that RandLoRA surpasses LoRA only when LoRA begins to overfit as the parameter count increases. I recommend the authors to qualify their statements to reflect that RandLoRA's superiority emerges prominently under conditions of LoRA’s overfitting.
(2): In the related work section of this paper, the authors have omitted some significant recent advancements in LoRA modifications. For example: SVFT, HydraLoRA, PISSA, LoRA-XS, FLoRA, etc. The inclusion of these advancements is essential for enriching the research background and understanding the current research progress in this field. While DoRA is mentioned in the related work, it is not compared with RandLoRA in the experimental section. I recommend that the authors consider such comparisons in future work. This would not only enhance the persuasiveness of the paper but also better showcase the advantages and distinct characteristics of RandLoRA among the plethora of methods.

---

> ### Author Response · Authors · 2024-11-20
>
> We thank the reviewer for their detailed review.
>
> - Q1: VeRA already pointed out the performance saturation as the rank of LoRA increases.
>
> Thank you for bringing this to our attention, we have added a citation to VeRA line 46.
>
> To the best of our knowledge, while VeRA noted that LoRA's parameter efficiency decreases with increasing ranks (Section 3.2, Paragraph 1 in the VeRA paper), it does not identify low rank as an inherent limitation. VeRA primarily focuses on minimizing trainable parameter count beyond rank-1 LoRAs (Section 2, Paragraph 2 and Contributions, Page 2, Point 1 in the VeRA paper) and on reducing disk storage requirements for fine-tuned weights (Section 3.2, Paragraph 2 in the VeRA paper).
>
> Our work identifies low-rank approximation as a limitation and proposes to decouple it from our parameter-efficient fine-tuning formulation. We propose a full-rank formulation, while maintaining the same parameter and memory efficiency. This allows our method to combine parameter-efficiency and full-rank to achieve strong performance.
>
> - Q2: RandLoRA is longer to train than LoRA, what are possibilities for improvement ?
>
> We acknowledged the reviewer's observation (L547-555) regarding RandLoRA's primary limitation: increased training time proportional to model size. To mitigate this, future enhancements would focus on implementing matmul-free matrix combinations as an efficient use of the ternary sparse random bases introduced in Section 6.4.
>
> Notably, our experiments (Table 4) demonstrate that sparse random matrices containing {-1, 0, 1} values maintain performance. An efficient implementation would thus simplify the matrix product of $B$ by $\Lambda A\Gamma$ to simple aggregations, eliminating floating-point arithmetic [1]. Although CUDA kernels for such operations are currently unavailable [2], their future development would significantly accelerate RandLoRA training. This optimization would alleviate RandLoRA's dominant limitations and we hope it inspires further community contributions.
>
> We revised Section 6.6's first paragraph to incorporate this discussion on matmul-free approaches. Additional strategies for overcoming RandLoRA's limitations are discussed in Section 6.6, including faster convergence via optimal random bases.
>
> - Q3: RandLoRA becomes more accurate for larger amounts trainable parameters but LoRA is still competitive for smaller parameter counts.
>
> We concur with the reviewer's assessment. To address their suggestion, we have explicitly clarified Line 74 that while LoRA can be preferable for scenarios where very high parameters-efficiency is required, RandLoRA presents an alternative in situations demanding larger parameter capacities. This typically includes challenging tasks or scenarios where available zero-shot models require significant adjustments.
>
> [1] Li, Ping, Trevor J. Hastie, and Kenneth W. Church. "Very sparse random projections."ACM SIGKDD international conference on Knowledge discovery and data mining. 2006.
>
> [2] Zhu, Rui-Jie, et al. "Scalable MatMul-free Language Modeling." arXiv preprint arXiv:2406.02528 (2024).

---

> ### Author Response · Authors · 2024-11-20
>
> - Q4: Comparisons with further related works that propose improvements over LoRA.
>
> We agree that an exhaustive comparison is desirable to improve understanding of the state of the field and promote fairer research.
>
> **Additions to Section 2.1**
>
> We have added the proposed recent works to Section 2.1 and point out that they showcase how important the parameter-efficient field is to the community. We kindly point out that to the best of our knowledge, only DoRA, FLoRA and SVFT have appeared in peer-reviewed conferences at this time.
>
> **Relation between RandLoRA and the suggested works**
>
> We note that RandLoRA's uniqueness, expressed through its full-rank parameter-efficient updates make it largely orthogonal to the suggested works. RandLoRA should then be considered more as an add-on on top of existing methods build around LoRA than a strict alternative.
>
> For instance, DoRA's update is represented as $m \times \frac{LoRA}{\| LoRA \|}$ (i.e. normalizing then scaling the updates), implying RandDoRA would naturally extend to $m \times \frac{RandLoRA}{\| RandLoRA \|}$. The multiple upscaling matrices of Hydra-LoRA could be computed using RandLoRA's strategy. LoRA-XS or SVFT could be used as alternatives to estimate SVD slices in Equation 6.
>
> Generally, we propose that novel methods that enhance LoRA's convergence without altering its rank can reasonably be expected to yield similar improvements in Rand-X variants.
>
> **Further comments on the related works**
>
> We identify that Pissa and DoRA are particularly relevant to the comparison as similarly to RandLoRA they propose improved results over LoRA for the same parameter count.
>
> Although relevant to the understanding of the research landscape, the other algorithms are less suited to the comparison as they serve goals miss-aligned to RandLoRA's: SVFT and LoRA-XS aim to greatly reduce the amount of parameters in LoRA with minimum decreases in accuracy, FLoRA is not parameter-efficient as it focuses on gradient rather than weight compression and Hydra-LoRA learns non-mergeable weights selected through a Mixture-of-Expert strategy, thus creating a new model at inference time.
>
> **Experimental results**
>
> For completeness of the experiments and following the reviewer's suggestion, we are now working on reporting DoRA results for the commonsense reasoning tasks on LLama3-8b in the newly-added Table 2.
>
> We will continue to update Table 2 with DoRA results as they come in to facilitate direct comparison and provide a clearer understanding of RandLoRA's position within the research landscape. We are also aiming to provide RandDoRA results by the camera-ready to support our point about compatibility.
>
> We thank the reviewer again for their in-depth review. Please let us know if you have further concerns you would like addressed.

---

> > ### Comment · Reviewer_ZGK2 · 2024-11-25
> > **After reading the author's response, I am willing to raise the score by one point.**
> >
> > After reading the author's response, I am willing to raise the score by one point.

---

### Official Review · Reviewer_rgkh · 2024-11-03

**Soundness:** 3
**Presentation:** 2
**Contribution:** 3
**Rating:** 6
**Confidence:** 4

**Summary:**

This paper proposes RandLoRA, a new method to address the limitations of LoRA in complex tasks. RandLoRA overcomes the low-rank constraint of LoRA by learning a combination of random low-rank basis matrices to achieve full-rank updates, and trend-off a balance between parameter efficiency and model performance. However, the paper needs to be further strengthened in several aspects. Overall, the paper is novel, but there is room for improvement in experimental and theoretical.

**Strengths:**

1. RandLoRA proposes a full-rank optimization strategy based on random low-rank matrix combinations, which helped the limitations of LoRA in complex tasks, especially the problem that its low-rank matrix cannot fully capture the complexity distribution of the task.
2. In the case of limited parameters, RandLoRA shows higher performance than LoRA, especially in vision-language tasks, showing that the method has certain parameter efficiency.
3. The paper provides novel ideas for the fine-tuning of large models, while reducing computing resource consumption and memory usage, while improving the performance of the model on specific tasks.
4. This paper well analysis the rationale behind the effectiveness of the proposed method.

**Weaknesses:**

1. The paper's derivation of RandLoRA is based on SVD and random basis matrix combination, but the theoretical rigour is still insufficient. The derivation assumes that the basis matrix obeys a specific random distribution (such as Gaussian or uniform distribution), which is difficult to strictly guarantee in practice. In addition, the combination of random basis matrices may cause stability problems in large-scale training. It is recommended to conduct experiments on models with larger parameter amounts to verify the robustness of the method.
2. Theorem 4.1 proposed in the paper gives the approximation error bound of RandLoRA, but does not explain in detail how to control the size of the error in practical applications, especially as the model size increases, whether the error will accumulate, which may affect its approximation effect.
3. The introduction of sparse matrices is intended to reduce computational complexity, but the impact of sparse matrices on the full-rank approximation effect has not been fully demonstrated. Although Table 3 shows the experimental effect of sparse matrices in RandLoRA, the paper does not explore the theoretical impact of sparse matrices in full-rank approximation in depth, and it is recommended to add analysis in this regard.
4. The comparative experiment of the paper selected LoRA, NoLA, VeRA and other parameter efficient fine-tuning methods, but did not include full parameter fine-tuning as a control. It may not be sufficient to select only LoRA as the main benchmark. It is recommended to supplement the full parameter fine-tuning results to fully evaluate the advantages and disadvantages of RandLoRA.
5. RandLoRA has relatively small improvements in visual tasks, but its effect in visual-language tasks is significantly enhanced. It may be related to the complexity of the task and the characteristics of multimodal data?
6. The impact of different configurations of RandLoRA (such as the sparsity of the random basis matrix and the distribution selection of the basis matrix) on the effect deserves further study. It is recommended to add ablation experiments on factors such as the basis matrix generation method and parameter scale to more comprehensively reveal the performance influencing factors of RandLoRA.
7. Although RandLoRA performs well on small-scale parameter models, its effectiveness in larger-scale models (such as LLaMA 70B and LlaVA 32B) has not been verified. It is recommended to conduct experiments on larger-scale models.

**Questions:**

see the weaknesses.

---

> ### Author Response · Authors · 2024-11-20
>
> We thank the reviewer for their review of our work.
>
> - Q1:  How can we guarantee that the random matrices follows a Gaussian or uniform distribution ?
>
> Thank you for raising concerns about potential limitations. We agree that is it practically very difficult to enforce a matrix to be strictly random following a particular distribution (such as Gaussian or Uniform). We use constructions from the torch library such as torch.nn.init.kaiming_uniform to generate pseudo-random matrices which is close to the best one can expect to achieve given that true randomness is unattainable in computational environments.
>
> - Q2: How can we guarantee there are no training instabilities for larger scale models ?
>
> Regarding potential instabilities, our experiments scaling up to 8B parameters with LLama3 have not revealed any training instabilities.
> Could you kindly elaborate on the potential instability risks associated with larger models the reviewer is referring to ?
>
> - Q3: How does the approximation error proposed in Theorem 4.1 behave with model size ?
>
> As model size increases, the number of values to estimate in $W$ grow quadratically, while trainable parameters in $\Lambda$ and $\Gamma$ increase linearly (similarly to LoRA or VeRA for example). All else being equal, this could theoretically suggests increasing approximation errors with larger models. We however point out that experiments with architectures up to 8B LLama3 show that this phenomenon is absent and that RandLoRA maintains performance.
>
> An important detail invalidating this intuition is that $W$ is a fine-tuned update on top of pre-trained weights $W_0$ that tend to become more expressive with increased model size and thus requiring simpler weight updates [1,2]. This would intuitively explain why $\epsilon$ (equation 6) remains small in practice. We observe this phenomenon in CLIP experiments, where VeRA's competitiveness improves with larger ViT-H/14 models (Figure 3). This derivation together with experimental results of Table 1 support RandLoRA's suitability for large model applications.
>
> In the hypothetical scenario where an accuracy degradations occurs, users can adjust the slice size r to estimate smaller, more manageable portions of W, thus reducing $\epsilon$. This adaptability parallels LoRA's adjustment of the rank r for harder problems.
>
> - Q4: Do sparse matrices preserve the full rank constraint ?
>
> The main concern for ternary sparse matrices could come from drawing the same row in the B matrices more than once over $N$ tries which would lead to a co-linearity and thus non-full rank. If we can show that the probability of this even happening is reasonably small then the full rank constraint would be preserved in practice.
>
> Given we draw $N$ matrices of size $D\times d$, this problem simplifies to $N\times D$ tries at drawing the same or exact opposite sequence of length $d$. Using the probability assignments from L480-481, we calculate that the probability of two rows being the same or opposite is $2\times (\frac{1}{2})^{d}= 1\times 10^{-231}$ for the smallest matrix we train on: ViT-B-32 of size $768\times 768$. Given N = 100, a fair estimate of a common configuration for RandLoRA, the probability of having any same or opposite sequence over $(N\times D)$ tries is $2\times (N\times D)(N\times D - 1)(\frac{1}{2})^{d}= 8\times 10^{-222}$ which enables sparse matrices to preserve the full rank in practice.
>
> We have added this clarifying comment at the end of section 6.4.
>
> [1] Hu, Edward J., et al. "Lora: Low-rank adaptation of large language models." ICLR 2022.
>
> [2] Aghajanyan, Armen, Luke Zettlemoyer, and Sonal Gupta. "Intrinsic dimensionality explains the effectiveness of language model fine-tuning." ACL 2021.

---

> > ### Comment · Reviewer_rgkh · 2024-12-02
> > **keep my original rating**
> >
> > the authors addrssed most of my concerns, and I am generally happy with the revision. I would like to keep my original rating.

---

> ### Author Response · Authors · 2024-11-20
>
> - Q5: Full fine-tuning as a baseline for language experiments.
>
> A Full-finetuning (FT) is included for as a baseline for vision and vision-language experiments in Figures 2, 3 and Tables 8-11 in the supplementary material. We omitted this baseline for the language experiments due to VRAM constraints on LLama3-8b particularly.
>
> We however report here a baseline for the smaller Phi3 network on the commonsense reasoning task where we find that full fine-tuning performs worse than parameter-efficient methods especially as the amount of training data increases. This most likely a result of over-fitting.
>
> We expect the same behavior for larger architectures such as LLama3-8b and refer to GPT-3 results in the original LoRA paper where LoRA performs better than full finetuning for these larger models (Table 4 in the original LoRA paper [1]). In conclusion, we use LoRA as our baseline because it performs better than full fine-tuning for large language models.
>
> | Method name | % Params | BoolQ | PIQA | SIQA | HellaSwag | WinoGrande | ARC-e | ARC-c | OBQA | Average |
> | --- | --- | --- | --- | --- | --- | --- | --- | --- | --- | --- |
> | Phi3 - 15k|
> | FT |  100.0 | 68.62 | 85.47 | 76.46 | 72.73 | 77.51 | 95.58 | 86.01 | 87.20 | 81.20 |
> | VeRA1024 | 0.015 | 68.53 | 84.49 | 73.08 | 74.54 | 72.85 | 93.01 | 80.97 | 81.60 | 78.63 |
> | LoRA-64 | 2.28 | 69.88 | 85.75 | 74.97 | 74.45 | 75.30 | 95.54 | 87.12 | 88.00 | 81.37 |
> | RandLoRA-10 | 2.29 | 69.63 | 85.31 | 75.03 | 86.94 | 75.30 | 95.24 | 85.58 | 86.40 | 82.43 |
> | Phi3 - 170k|
> | FT | 100.0 | 71.62 | 87.60 | 79.22 | 59.97 | 83.74 | 95.33 | 86.35 | 90.00 | 81.73 |
> | VeRA1024 | 0.015 | 69.53 | 84.53 | 74.52 | 84.08 | 76.82 | 94.51 | 83.68 | 83.54 | 81.40 |
> | LoRA-64 | 2.28 | 71.93 | 86.13 | 79.58 | 90.14 | 83.74 | 92.68 | 81.74 | 87.80 | 84.22 |
> | RandLoRA-10 | 2.29 | 71.87 | 86.56 | 79.43 | 90.99 | 82.72 | 95.66 | 85.49 | 87.40 | 85.01|
>
> - Q6: Are the larger improvements on Vision language due to increased complexity ?
>
> Yes we concur with the analysis of the reviewer on the increase in complexity of the task, probably as well due to the dual backbone nature of the vision-language architectures. We added a line at the end of Section 5.3 using this insight to justify the improvements on Vision-language tasks.
>
> - Q7: More ablations on sparse random bases and type of random base distributions deserve further study.
>
> We provide early results on the effect of sparse bases in Table 3 where we observe a small decrease in accuracy results over the widely used CLIP and LLama3 architectures. We expect these results to generalize to other architectures as well.
>
> Regarding the ablations on the distribution the random bases are drawn from, we refer the reviewer the VeRA paper which performed such studies (Table 6 and 7 in the VeRA paper). Because the VeRA formulation is currently the base for our sliced SVD approximation (equation 6), we used VeRA's design choices regarding random base creation and expect their ablation results to hold for RandLoRA.
>
> Since VeRA did not propose to use sparse bases however, we are now working on providing ablation experiments using random bases with different levels of sparsity [3] as well as Gaussian bases to add to Table 4. Early results indicate slight decreases of accuracy with sparser bases and Gaussian bases. We will report the full results as soon as they become available.
>
> - Q8: Run RandLoRA on LLama-70B and LLava-35B.
>
> We acknowledge and share your interest in exploring RandLoRA's performance on extreme model sizes. Although we haven't reported results on these very large architectures, Figure 3 and Table 1 demonstrate a promising trend: RandLoRA consistently improves upon LoRA as network size increases, suggesting scalability to even larger architectures.
>
> We have attempted to train a LoRA baseline on a quantized version of LLama-70B on 4xA100 GPUs and calculated it would require a 6 days runtime, inference and hyper-parameter search excluded. Given the two-week response period and our limited compute resources, fine-tuning these architectures in time is unfortunately not realizable.
>
> [1] Hu, Edward J., et al. "Lora: Low-rank adaptation of large language models." ICLR 2022.
>
> [3] Li, Ping, Trevor J. Hastie, and Kenneth W. Church. "Very sparse random projections." ACM SIGKDD international conference on Knowledge discovery and data mining. 2006.
>
> We thank the reviewer again for their detailed review. Please let us know if you have further concerns you would like addressed.

---

> ### Author Response · Authors · 2024-11-28
>
> Dear reviewer rgkh,
>
> We have now obtained the results for the suggest ablation study of the random bases. We have studied normally distributed and uniform binary random bases as well as sparser variations for the sparse bases.
> We find that by following the maximum sparsity suggestion of Li et al [3] we obtain 93\% sparse bases that maintain close performance to the dense random bases. When pushing above this limit to 98\% or 99\% sparsity we observe significant degradations in performance.
>
> | Model | Sparsity | Accuracy |
> | --- | --- | --- |
> | CLIP-ViT-B/32 - uniform | 0% | 85.98 |
> | CLIP-ViT-B/32 - normal | 0% | 85.61 |
> | CLIP-ViT-B/32 - binary | 0% | 85.52 |
> | CLIP-ViT-B/32 | 66% | 85.43 |
> | CLIP-ViT-B/32 | 93% | 85.57 |
> | CLIP-ViT-B/32 | 98% | 84.35 |
> | CLIP-ViT-B/32 | 99% | 83.34 |
>
> We have added these results to Table 4 and discussed them line 524-530 in Section 6.4 of the updated manuscript (blue text). We are additionally working on evaluating the 93\% sparse bases for LLama3 and we report the results soon.
>
> The authors

---

> ### Comment · Area_Chair_s6jV · 2024-11-30
>
> Dear Reviewer,
>
> Could you kindly respond and indicate whether authors have addressed your concerns?
>
> Thanks, AC

---

### Official Review · Reviewer_ufJr · 2024-11-04

**Soundness:** 2
**Presentation:** 3
**Contribution:** 2
**Rating:** 6
**Confidence:** 4

**Summary:**

The paper proposes RandLoRA for parameter-efficient fine-tuning for vision and language models. The authors start from analyzing the drawbacks of traditional low rank adaptation methods(LoRA) and argue that the importance of non-essential ranks during adaptation. RandLoRA shows better performance than existing methods on fine-tuning CLIP models on image classification and fine-tuning LLMs on 8 commonsense reasoning tasks.

**Strengths:**

- The presentation is clear and easy to understand
- The proposed RandLoRA's convergence has been theoretically proved
- Various experiments on different tasks and models are done

**Weaknesses:**

- Limited technical novelty. What is the main difference between VeRA and RandLoRA? There is a fairly similar update formulation in VeRA, e.g. two frozen low-rank matrices and two trainable small matrices.
- Lack of some important experiments for further verification. Most competitors, e.g. VeRA and LoRA, in the paper are proposed for language models and language tasks. To confirm the superiority of RandLoRA, the authors should directly compare the performance between RandLoRA and former competitors on standard language tasks, e.g. GLUE and E2E used in VeRA.

**Questions:**

See weaknesses.

---

> ### Author Response · Authors · 2024-11-20
>
> We thank the reviewer for the detailed review of our work.
>
> - Q1: What is the difference between VeRA and RandLoRA ?
>
> A discussion is available in Section 6.5 which we detail further here.
>
> RandLoRA and VeRA stem from distinct hypotheses regarding the optimal rank of weight updates when fine-tuning large pre-trained model. VeRA adopts LoRA's low-rank approximation, while RandLoRA targets full-rank updates. Essentially, VeRA efficiently approximates LoRA, whereas RandLoRA explicitly identifies non-full ranks as the limit to the quality of the update over the amount of trainable parameters.
>
> For context, RandLoRA sums over all rank-r sections of the full-rank weight update's SVD decomposition (Section 4.2) by leveraging parameter-efficient estimations of each section. Similarities with VeRA here stem from the empirical confirmation that VeRA's formulation is effective at estimating these low-rank sections. In this context, RandLoRA's framework allows for future improvements to SVD section approximations, such as other formulations different to VeRA's that further reduce in Equation 6 or by using hybrid estimations such as explicit estimation of critical sections using LoRA (lines 558-565).
>
> We further clarify here that RandLoRA's current solution is not simply a sum of multiple VeRAs since: we propose the use of a single random base A to constrain memory use (equation 4), demonstrate the applicability of sparse random bases to reduce compute requirements (section 6.4) and derive theoretical bounds to justify convergence (Theorem 4.1).
> Finally VeRA is capped by definition at (D+r) trainable parameters for weight matrices whereas RandLoRA accommodates for larger budgets (line 233).
>
> To conclude, VeRA and RandLoRA cater to different needs of the parameter-efficient community. While VeRA enhances LoRA's parameter efficiency for ultra-small parameter budgets, RandLoRA adresses tasks requireing moderately larger budgets (Figure 1).
>
> - Q2 GLUE and E2E results for RandLoRA
>
> We appreciate your suggestion to include GLUE and E2E results for comprehensive comparison which we are now working on providing.
> In the interim, we refer the reviewer to Section 5.4 and Table 1, showcasing RandLoRA's enhancements over the LoRA and VeRA baselines in challenging commonsense reasoning language tasks. Notably, RandLoRA demonstrates good improvements with the LLama3-8B architecture which is highly relevant to the research community.
>
> Regarding GLUE, we anticipate modest improvements of RandLoRA over LoRA due to three factors: 1) GLUE tasks primarily involve simple binary classification, 2) LoRA already bridges the gap with full fine-tuning, and 3) VeRA reports performances on par with LoRA's.
>
> We have provided results for GLUE using the RoBERTa-base architecture (125M parameters), averaged over 5 runs. As expected, the results for each algorithm are very close and not statistically different. We will make the results for the RoBERTa-large architecture available as soon as they come in with early trends pointing to similar results to the RoBERTa-base architecture. We are additionally currently running RandLoRA on E2E with GPT2-medium.
>
> Results on GLUE datasets with RoBERTa-base. We report the same metrics as in the VeRA paper: Matthew’s correlation for CoLA, Pearson correlation for STS-B, and accuracy for the remaining tasks. As VeRA did not release their code, we use hugginface's transformers official implementation: https://github.com/huggingface/transformers/tree/main/examples/pytorch/text-classification . Contrary to VeRA and to save on large search times, we use the same hyper-parameters across all tasks. All algorithms are run using the same code base.
>
> | Method | Params | SST-2 | MRPC | COLA | QNLI | RTE | STS-N | Average |
> | --- | --- | --- | --- | --- | --- | --- | --- | --- |
> LoRA-4| 0.7M | 94.4 ± 0.5 | 87.3 ± 0.2 | 58.4 ± 0.8 | 92.7 ± 0.2 | 71.5 ± 1.2 | 90.5 ± 0.1 | 82.4 ± 0.3|
> RandLoRA-64| 0.7M | 92.2 ± 0.3 | 88.0 ± 1.5 | 59.4 ± 2.1 | 91.3 ± 0.4 | 74.7 ± 1.9 | 90.3 ± 0.2 | 82.6 ± 0.5|
> VeRA-1024| 0.2M | 91.9 ± 0.4 | 88.4 ± 1.2 | 59.9 ± 2.2 | 90.5 ± 0.4 | 74.9 ± 1.5 | 90.4 ± 0.2 | 82.7 ± 0.3 |
>
> Thank you again for your review, we will share further results here as they become available. Please let us know if you have remaining concerns you would like addressed.

---

> ### Author Response · Authors · 2024-11-25
>
> Dear reviewer ufJr,
>
> Please find below further results.
> We have run the RoBERTa-large architecture on GLUE using the same settings as specified in our previous comment (5 runs).
>
> We find that in this case, RandLoRA improves over both LoRA and VeRA. These results indicate how RandLoRA becomes more beneficial for larger network sizes. This observation is in line with the results presented in Figure 3 where RandLoRA outperforms fine-tuning more with increased network sizes.
>
> | Method | Params | SST-2 | MRPC | COLA | QNLI | RTE | STS-N | Average |
> | --- | --- | --- | --- | --- | --- | --- | --- | --- |
> LoRA-4| 1.8M | 95.5 ± 0.2 | 87.2 ± 0.7 | 64.7 ± 1.2 | 94.5 ± 0.1 | 83.6 ± 0.4 | 91.8 ± 0.1 | 86.2 ± 0.3 |
> RandLoRA-100| 1.8M | 95.5 ± 0.3 | 90.1 ± 0.4 | 67.4 ± 0.3 | 94.1 ± 0.3 | 84.5 ± 0.3 | 91.4 ± 0.6 | 87.2 ± 0.1 |
> VeRA-256| 0.26M | 95.8 ± 0.3 | 89.3 ± 1.2 | 65.3 ± 1.1 | 94.1 ± 0.3 | 81.6 ± 0.8 | 91.8 ± 0.1 | 86.3 ± 0.3 |
>
> We additionally report here results on the E2E dataset with GPT2-Medium, we use LoRA's official implementation and the recommended hyper-parameters. We find that RandLoRA slightly outperforms LoRA for the BLEU and NIST scores. We are still working on reproducing VeRA's results.
>
> | Method | BLEU | NIST | METEOR | ROUGE_L | CIDEr |
> | --- | --- | --- | --- | --- | --- |
> LoRA-16 | 68.4 | 8.62 | 46.3 | 71.4 | 2.5 |
> RandLoRA-20 | 69.0 | 8.71 | 46.3 | 71.4 | 2.5 |
>
> The authors

---

> ### Comment · Reviewer_ufJr · 2024-11-28
> **Response to the rebuttal**
>
> Thank you for the rebuttal. Most of my concerns are solved. I'm willing to raise my score.

---

### Official Review · Reviewer_uANM · 2024-11-04

**Soundness:** 2
**Presentation:** 3
**Contribution:** 2
**Rating:** 6
**Confidence:** 4

**Summary:**

This paper introduces RandLoRA, a method designed for efficient parameter tuning of both visual and linguistic models. The researchers discuss the shortcomings of conventional low-rank adaptation techniques, known as LoRA, and highlight the significance of non-critical ranks in the adaptation process. As a result, compared with traditional LoRA, RandLoRA earns better performance with fewer trainable parameters. The convergence of RandLoRA is discussed in detail. Extensive experiments verify its effectiveness on vision and language tasks.

**Strengths:**

+ RandLoRA is proposed to approximate low-rank updates under a clear motivation about the importance of non-critical ranks.
+ Multiple scales of models are selected as baselines, and RandLoRA can lead to good improvement in most situations.
+ The paper is well-written and easy to follow.

**Weaknesses:**

- The motivation here mainly focus on how to approximate and improve low-rank adaptation methods like LoRA. The conclusion is to use full-rank updates and thus the authors propose RandLoRA. However, RandLoRA also outperforms full fine-tuning in various tasks like image classification. How to explain this experimental result? Why we can earn improvement by approximating low-rank updates to full-rank updates over both LoRA and full fine-tuning?
- Some important baselines are missing. For example, in the field of tuning CLIP on image classification tasks, many state-of-the-art methods use prompt-based tuning methods, e.g. PromptSRC (ICCV'23)[a], DePT (CVPR'24)[b] instead of LoRA. Such kind of parameter-efficient fine-tuning methods should also be discussed and compared with, given that the most-related works VeRA and LoRA are not initially proposed for image classification tasks.

[a] Self-regulating Prompts: Foundational Model Adaptation without Forgetting, https://arxiv.org/abs/2307.06948)
[b] DePT: Decoupled Prompt Tuning, https://arxiv.org/abs/2309.07439

**Questions:**

Please see weakness.

---

> ### Author Response · Authors · 2024-11-20
>
> We thank the reviewer for the detailed review of our work.
>
> - Q1: Why does RandLoRA outperform fine-tuning in various tasks like image classification?
>
> RandLoRA indeed improves performance over fine-tuning in some cases, especially for vision-and-language models such as CLIP, as shown in Figure 3 where RandLoRA outperforms fine-tuning despite having fewer trainable parameters. We find that in these cases, full fine-tuning is highly susceptible to over-fitting, thus compromising generalization accuracy. In contrast, RandLoRA optimizes over a lower-dimensional parameter space, which mitigates over-fitting and leads to better generalization in this case. This is corroborated by the observation that RandLoRA outperforms fine-tuning more on larger models in Figure 3.
>
> The superior performance of low-rank methods when fine-tuning language models is also observed in the original paper, where LoRA [1] is reported to sometimes outperform full fine-tuning. This is particularly evident in the GPT-3 experiments (Table 4 in the LoRA paper [1]). We also evidence this phenomenon when full-finetuning Phi3 on the commonsense reasoning tasks. This is further detailed in the response to reviewer rgkh.
>
> To shed light on the overfitting, we would like to point out that in Figure 5 in the supplementary material, particularly in Figure 5b, full fine-tuning leads to much deeper training loss minima. These results highlight how that lowered sensitivity to over-fitting is the main reason behind RandLoRA's improvements over full-finetuning in CLIP or language architectures.
>
> - Q2: Given equal training parameters, why is it better to perform a full rank update.
>
> Full rank updates allow to explore any direction in the weight space. This is in contrast to LoRA which restricts exploring $r$ (rank) directions. In some cases such as vision-language (Figure 3) or commonsense reasoning (Table 1) we find that having access to all directions in the weight space become important to train accurate models and as a result RandLoRA outperforms LoRA.
>
> - Q3: RandLoRA should compare with prompt-tuning baselines which are specifically designed for few-shot image classification.
>
> Thank you for suggesting these related works, we have now included them in the new Section 2.3 to better contextualize our work in the parameter-efficient landscape.
>
> Although they tackle similar problems, we argue that these algorithms are orthogonal to our proposed method. Hence, RandLoRA should not be considered in direct competition but complementary. Probably for the same reason, the papers suggested by the reviewer also do not compare with LoRA. In general, while prompt-tuning is designed for few-shot settings exclusively, RandLoRA's competitiveness increases with moderately larger parameter budgets (Figure 1) and generalizes to full dataset fine-tuning.
>
> A comprehensive investigation into the performance of LoRA compared to prompt-tuning algorithms has however been recently been performed [2] where LoRA was found to be comparable to recent prompt-tuning alternatives for low-shot scenarios but largely outperformed for larger amounts of shots. This discussion has been summarized and included in Section 2.3 of the paper.
> We agree that further research demonstrating the complementarity of both approaches would be relevant to the community but it is currently beyond the scope of this work.
>
> We are however working now on running the PromptSRC+DePT configuration as an extension to our vision experiments to better contextualize our work in the parameter-efficient landscape and confirm the results of [2]. We will post the results here if they complete before the rebuttal deadline and include them in the camera ready.
>
> [1] Hu, Edward J., et al. "Lora: Low-rank adaptation of large language models." ICLR 2022.
>
> [2] Zanella, Maxime, and Ismail Ben Ayed. "Low-Rank Few-Shot Adaptation of Vision-Language Models." CVPR 2024.
>
> Thank you again for your review, please let us know if you have remaining concerns you would like addressed.

---

> ### Author Response · Authors · 2024-11-25
>
> Dear reviewer uANM,
>
> Prompt tuning experiments have now completed.
>
> We have been unable to run the PromptSRC+DePT configuration as the code for this configuration is not publicly released. The DePT paper reports Maple[3]+DePT as a close second best so we have chosen to report this configuration instead as it accurately reflect state-of-the-art performances for prompt tuning.
>
> We report results for 4 and 16-shots over the datasets selected in the DePT paper. We train CLIP in its ViT-B/32 variant with Maple[3]+DePT, RandLoRA-10 and LoRA-16 all training approximately 3M parameters.
>
> We find, as suggested in our previous comment, that while the Maple+Dept configuration compares favorably for the 4-shots setting, it struggles to keep up for 16-shots. We additionally report that Maple + DePT requires a much longer training time than both LoRA and RandLoRA, especially as the number of classes increases. For example, training 16-shots for 10 epochs on ImageNet requires 3.5h and 17GB of VRAM for Maple + DePT while it requires 2 minutes and 4.5GB of VRAM for RandLoRA.
>
>
> | 4-shots | ImageNet | Caltech101 | OxfordIIITPet | Cars | Flowers102 | Food101 | FGVCAircraft | SUN397 | DTD | EuroSAT | UCF101 | Average |
> | --- | --- | --- | --- | --- | --- | --- | --- | --- | --- | --- | --- | --- |
> LoRA-16| 64.9 | 92.0 | 88.2 | 63.9 | 87.9 | 82.6 | 30.3 | 68.2 | 61.1 | 89.4 | 74.7 | 73.0 |
> RandLoRA-10| 63.9 | 91.7 | 86.4 | 67.0 | 89.9 | 80.8 | 34.0 | 69.7 | 62.4 | 84.4 | 74.9 | 73.2 |
> Maple + DePT | 62.1 | 95.0 | 89.5 | 68.7 | 90.5 | 79.6 | 28.3 | 70.2 | 61.7 | 81.4 | 76.6 | 73.1 |
>
> | 16-shots | ImageNet | Caltech101 | OxfordIIITPet | Cars | Flowers102 | Food101 | FGVCAircraft | SUN397 | DTD | EuroSAT | UCF101 | Average |
> | --- | --- | --- | --- | --- | --- | --- | --- | --- | --- | --- | --- | --- |
> LoRA-16| 65.8 | 91.7 | 89.5 | 80.1 | 94.9 | 81.8 | 42.5 | 73.5 | 72.0 | 91.2 | 81.5 | 78.6 |
> RandLoRA-10| 66.3 | 95.6 | 91.1 | 77.4 | 94.5 | 84.0 | 45.0 | 73.7 | 72.5 | 94.1 | 81.7 | 79.6 |
> Maple + DePT | 67.7 | 96.0 | 90.5 | 79.1 | 96.3 | 81.7 | 36.9 | 74.5 | 70.3 | 90.3 | 82.1 | 78.7 |
>
> The authors
>
> [3] Khattak, Muhammad Uzair, et al. "Maple: Multi-modal prompt learning." Proceedings of the IEEE/CVF Conference on Computer Vision and Pattern Recognition. 2023.

---

> ### Comment · Area_Chair_s6jV · 2024-11-30
>
> Dear Reviewer,
>
> Could you kindly respond and indicate whether authors have addressed your concerns?
>
> Thanks, AC

---

> ### Comment · Reviewer_uANM · 2024-12-02
>
> Thank the authors for addressing most of my concerns. I will raise my score.

---

### Author Response · Authors · 2024-11-20

We thank all reviewers for their thoughtful comments and feedback on the limitations they evidenced in our work. We have given exhaustive answers to every questions to elicit an insightful discussion to allow us to address concerns and satisfy the acceptance threshold. We have updated the paper draft with the proposed changes and highlighted them in a blue color to facilitate review.

A large amount of additional experiments have been requested to further test the limits of RandLoRA or to add further comparisons with related work. We will do our best to report experimental results as they come in and hopefully in time given our academic resources.

Note that we have performed extensive experiments in our paper across multiple tasks and datasets: vision in section 5.2 and appendix D.2, vision-language in section 5.3 and appendix D.1, and commonsense-reasoning in section 5.3 and appendix D.3; where we ablate both number of trainable parameters of the PEFT methods and network sizes. These tests extensively report on RandLoRA's strengths and weaknesses and we believe give reliable trends toward generalization to larger architectures and other tasks.

In the meantime, please let us know if you require further clarifications or have additional concerns you would like addressed.

---

> ### Author Response · Authors · 2024-11-28
> **The authors invite the reviewers to voice any remaning concerns -- Manuscript update.**
>
> # Manuscript and supplementary material update
>
> We have updated the manuscript with further results according to the reviewer's suggestions.
>
> Results for **GLUE** and **E2E** benchmarks can be found in Table 5 in appendix B.1 and B.2 respectively. Results for **Maple+DePT** as a state-of-the-art prompt-tuning baseline is available in Table 6 in appendix B.3. We have referenced these additional results in the main manuscript lines 287-289.
>
> We have updated the ablation study of Table 4 with additional results using up to 99\% sparse bases. We are now working on applying the 93\% sparse bases to LLama3 to complement the 66\% sparse results that were initially available.
>
> Results for Table 2 comparing **RandLoRA with DoRA** are now available. We have used DoRA's official implementation and found that for commonsense reasoning, DoRA performs very similarly to LoRA. Furthermore we now report the training time for DoRA as well.
>
> We find that because DoRA's normalization strategy necessitates to explicitly compute the $BA$ update, it leads to a 2.2x increase in training time over LoRA with no obvious improvement paths. As a comparison, RandLoRA-30 leads to a 1.7x increase. We have included DoRA's training time for LLama3 in appendix C.6.1.
>
> # Only reviewer ZGK2 answered so far
>
> As the requested additional experiment have now completed, we invite the reviewers to reassess our work in light of these additional results and of our responses to their concerns. If their concerns have been addressed, we encourage the reviewers to share their revised evaluation and to consider increasing their acceptance score.
>
> The authors

---

### Comment · Area_Chair_s6jV · 2024-11-28
**Reviewers, please kindly respond**

Dear Reviewers,

If you have not responded to author's rebuttal, please kindly do so as soon as possible. The deadline is Dec 2, but the authors can potentially further clarify questions if you respond earlier. Thanks!

Best, AC

---

### Meta-Review · Area_Chair_s6jV · 2024-12-24

**Metareview:**

Summary: a new PEFT method that uses parameters to combine random matrices as weight updates; it demonstrates ranks are important, and with minimal parameters, it can achieve competitive performance on various tasks.

Strengths: simple yet effective full-rank PEFT method; extensive experiments on different modalities; clear presentation; outperforms LoRA.

Weaknesses: increased training time; some missing baselines; no large-scale models (>30B).

Reason for decisions: demonstrated the importance of rank; a simple and effective method; all reviewers are leaning positive.

**Additional Comments On Reviewer Discussion:**

The authors addressed many concerns, adding experiments (e.g., GLUE, E2E, and sparse bases), clarifying random bases' properties, and refining theoretical analysis.

---

### Decision · Program_Chairs · 2025-01-22

Accept (Poster)